# Position: CNNs Don't See Shape — And That Won't Change Without New Architectures

**Ali Kayyam** [1]

## Abstract

Whether deep vision models recognize objects primarily by shape or texture remains a central and unresolved question in computer vision. Early studies report a strong texture bias in convolutional neural networks (CNNs), while other work reports shape-biased representations. We argue that much of this apparent discrepancy reflects methodological confounds and a conflation of local contour sensitivity with genuine global shape understanding. Using minimal, tightly controlled stimuli, we directly compare cue-conflict and cue-suppression paradigms within a unified experimental framework. We show that standard CNNs consistently prioritize texture over global shape when cues compete, even when shape information is explicitly available. Evidence for shape bias typically reflects reliance on local fragments rather than invariant, relational representations of object structure. Our findings support the view that texture bias is fundamentally rooted in architectural inductive biases rather than data or optimization alone. This gap has direct consequences for robustness, safety, and generalization, and motivates the development of architectures that explicitly support global integration and relational reasoning, moving beyond incremental data-driven fixes. The code is publicly available at https://github.com/alikayyam/shape_vs_texture.

## 1. Introduction

Despite their impressive performance on large-scale benchmarks, deep neural networks (DNNs) for vision still lag behind humans in their ability to generalize to novel input distributions. A growing body of evidence suggests that this gap arises not from insufficient capacity, but from differences in the visual features prioritized by models and humans. In particular, CNNs tend to rely on local **texture** cues, whereas human vision predominantly uses **global shape** information for object recognition. Please see Figs. 5 and 6.

The influential work of Geirhos et al. (2018a) formalized this observation by showing that CNNs classify objects based on texture when shape and texture cues conflict. This finding has motivated extensive follow-up work proposing architectural changes, training schemes, and evaluation protocols to induce more human-like, shape-based representations (Baker et al., 2018; Hermann & Lampinen, 2020). However, the degree to which CNNs truly learn shape—rather than exploiting alternative local cues—remains a subject of active debate.

From an architectural perspective, texture bias is a natural consequence of the **local inductive bias of convolution**. Convolutional filters operate over small receptive fields and efficiently extract low-level features such as edges and gradients, which readily compose into textures. Further, pooling operations discard precise spatial relationships critical to shape. In high-dimensional natural image datasets, texture often provides a statistically reliable and low-complexity shortcut for classification (Hermann & Lampinen, 2020). In contrast, shape perception requires integrating information across extended spatial regions and capturing relational structure, making it a fundamentally harder learning problem. While deeper networks increase receptive field size, this does not guarantee global reasoning; CNNs often function as aggregations of local evidence rather than holistic shape processors.

Empirically, CNNs can classify heavily texture-scrambled images (Brendel & Bethge, 2019) (Fig. 17), and models restricted to very small local patches (e.g. BagNets) achieve near state-of-the-art accuracy, indicating that global shape is often unnecessary for correct classification. Conversely, CNNs perform poorly on shape-dominant stimuli such as silhouettes, line drawings, or contour-only images unless trained with specialized procedures (Baker et al., 2018). These failures contrast sharply with human performance and are closely linked to practical shortcomings, including vulnerability to out-of-distribution shifts (Geirhos et al., 2018b), adversarial perturbations (Szegedy et al., 2013),

[1]BrainChip Inc., Laguna Hills, CA, USA. Correspondence to: Ali Kayyam <akayyam@brainchip.com>.

*Proceedings of the 43$^{rd}$ International Conference on Machine Learning*, Seoul, South Korea. PMLR 306, 2026. Copyright 2026 by the author(s).

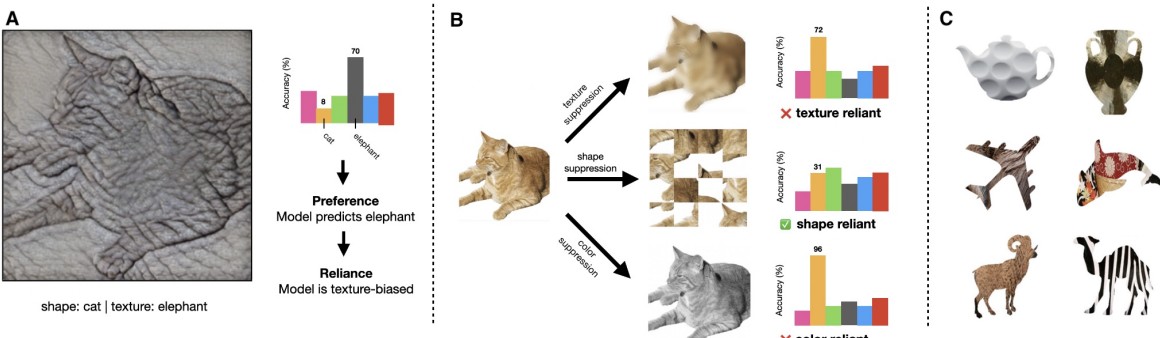

*Figure 1.* Comparison of two common experimental paradigms used to probe shape versus texture bias in CNNs. **(A)** Cue-conflict paradigm introduced by Geirhos et al. (2018a), in which object shape and texture are deliberately placed in conflict. A prediction consistent with texture (e.g. "elephant") is interpreted as evidence of texture bias, whereas a prediction consistent with shape (e.g. "cat") indicates shape bias. A key limitation of the original cue-conflict setup is that texture is applied not only to the object but also to the background, creating unnatural scenes and introducing additional cues that models may exploit. See also Fig. 16 in Appendix. Baker et al. (2018) addressed this issue by restricting texture manipulation to the object region only (panel C). **(B)** Cue-suppression paradigm used by Burgert et al. (2025), which attempts to selectively remove either shape or texture cues. As can be seen, cue-suppression approaches suffer from complementary limitations: neither shape nor texture cues can be completely eliminated, leaking residual information that may confound interpretation. **(C)** Experimental design proposed by Baker et al. (2018). It avoids background texture confounds and minimizes residual cue leakage. We analyze and discuss different variations of these design paradigms using simple and controlled stimuli.

and reduced robustness to common image corruptions.

Several studies have shown that shape sensitivity can be increased through data augmentation or stylized training that suppresses texture cues (Geirhos et al., 2018a; Hermann & Lampinen, 2020; He et al., 2023b; Lee et al., 2022; Tripathi et al., 2023; Ge et al., 2022). While such approaches improve robustness and increase measured shape bias, they do not resolve whether CNNs *prefer* shape or merely rely on it once texture shortcuts are removed. Notably, even under these conditions, models remain substantially less shape-biased than humans, suggesting that texture bias is reduced statistically rather than eliminated architecturally.

More recent work (Burgert et al., 2025), in contrast to Geirhos et al. (2018a), argues that CNNs do exploit shape information and exhibit apparent shape bias when evaluated using alternative paradigms, such as cue suppression. We argue that many apparent contradictions in the literature arise from methodological differences: whether experiments rely on synthetic stimuli, suppress rather than disentangle cues, or probe local contour fragments rather than global object structure. Each approach captures a partial view of shape processing, and evidence for shape bias often reflects sensitivity to contour fragments rather than invariant, relational shape representations.

We critically review prior experimental paradigms, outlining their respective strengths and limitations, and then demonstrate—using a simple and controlled design—that the conclusions drawn from the approaches of Geirhos et al. (2018a) and Burgert et al. (2025) are not inherently contradictory (Fig. 1). We take the position that **purely feed-forward convolutional networks remain fundamentally texture-biased**. While shape information may be present implicitly

and can be amplified through large-scale training or specialized data regimes, robust, human-like shape perception does not emerge naturally from standard CNN architectures. We argue that resolving the shape–texture debate requires controlled experiments that distinguish local contour detection from true global shape integration. Clarifying this distinction is essential for building vision systems that generalize robustly and align more closely with human perception.

Burgert et al. (2025) do not argue that CNNs are globally shape-biased in the human sense, but instead suggest that CNN representations are better characterized by reliance on local shape fragments or patch-level structure, rather than pure texture statistics. In contrast, Geirhos et al. (2018a) place CNNs closer to the texture end of the spectrum based on cue-conflict stimuli. These views are not strictly contradictory; rather, they reflect different operationalizations of "shape." The central issue is the scale of spatial integration, and a conservative consensus is that CNNs, unlike humans, do not robustly encode or exploit global object shape.

## 2. Related Works
### 2.1. Evidence for Texture Bias in CNNs

The distinction between shape-based and texture-based object recognition in deep neural networks was most prominently articulated by Geirhos et al. (2018a), who demonstrated that ImageNet-trained CNNs overwhelmingly rely on local texture cues when global shape and texture are placed in conflict. This finding revealed a systematic divergence between human and model behavior and established shape bias as a diagnostic tool for probing robustness and generalization. Subsequent studies have largely confirmed the prevalence of texture bias across architectures and datasets,

*Table 1.* Summary of methods used to probe shape processing in deep networks. See also Fig. 19 in Appendix.

| Method Category | Core Idea | What It Tests | Key Strengths | Key Limitations |
|---|---|---|---|---|
| Cue-Conflict Paradigm (Baker et al., 2018; 2020; Hermann & Lampinen, 2020) | Generate images where shape and another cue are put in conflict (e.g. shape of one class + texture of another). | Which feature the network prioritizes in classification. | Tests relative feature biases directly. | Relies on classification outputs; doesn't distinguish local vs global shape; can be influenced by texture dominance. |
| Cue-Suppression (a.k.a., Diagnostic Stimuli) (Kubilius et al., 2016; Burgert et al., 2025) | Create or manipulate images so that only shape (or local vs global shape) information is available (e.g. silhouettes, scrambled shapes). | Whether networks can use shape cues, and whether they rely on local vs global shape information. | Isolates shape features; interpretable outcomes. | Often uses out-of-distribution stimuli far from training data; may not reflect natural image behavior. |
| Triplet Tests (Ritter et al., 2017; Guest & Love, 2019; Tartaglini et al., 2022; Malhotra et al., 2020) | Present a probe image plus two matches differing either in shape or another feature and compare representation similarity distances. | Whether representations prioritize shape over other features (shape bias) in layer activations. | Doesn't require classification decisions; can probe internal representations. | Sensitive to confounds like pixel similarity, position, object size; may reflect low-level similarity rather than true shape processing. |
| Representation Analysis (Kalfas et al., 2018; Kriegeskorte et al., 2008; Islam et al., 2021; Singer et al., 2022; Hermann & Lampinen, 2020) | Analyze network layers across many images with methods like decoding or representational similarity analysis (RSA). | Whether shape information is represented in intermediate layers and how that changes across layers. | Does not depend only on classification output; captures broader representational geometry. | Requires large image sets; interpretation of representational geometry can be complex. |

including AlexNet, VGG, and ResNet (Baker et al., 2018; Geirhos et al., 2018a; Hoak et al., 2025; Bowers et al., 2023; Jarvers & Neumann, 2023). It has been shown that architectures that perform better on ImageNet generally exhibit lower texture bias.

Architectural analyses further support this conclusion. Brendel & Bethge (2019) introduced BagNets, which operate solely on small local image patches, and showed that such models can achieve competitive ImageNet performance, indicating that global shape information is not required for high classification accuracy. More recent behavioral probes, including visual anagrams (Doshi et al., 2025), similarly reveal substantial differences in holistic shape processing across model families.

While some works report conditions under which networks appear more shape-biased (Ritter et al., 2017; Tartaglini et al., 2022), these findings often depend strongly on the evaluation protocol or training regime. Overall, the dominant pattern remains that CNN decisions are driven primarily by texture-like local statistics rather than global shape.

## 2.2. Evidence for Shape Bias in CNNs

A parallel line of work has demonstrated that CNNs do encode shape-related information in their internal representations. Using representational similarity analysis, triplet tests, and linear decoding, several studies have shown that intermediate layers cluster stimuli by contour similarity or allow above-chance decoding of shape attributes (Kubilius et al., 2016; Ritter et al., 2017; Islam et al., 2021). CNNs can successfully classify silhouettes or edge-based representations under controlled conditions (Baker et al., 2018).

However, this body of evidence primarily establishes shape sensitivity rather than shape dominance. Encoding shape information does not imply that such information causally drives model decisions when competing cues are available. Indeed, cue-conflict experiments consistently show that

shape information present in intermediate layers is often down-weighted or discarded in later decision stages, particularly in fully connected layers. This distinction between representational content and decision control is frequently blurred in the literature, leading to overly strong claims about human-like shape processing in CNNs.

## 2.3. Links Between Shape Bias and Robust Recognition

Incorporating shape-bias in CNNs (via stylized images, edge cues, etc; Fig. 13) has been shown to increase robustness to common distortions (noise, blur, etc.) (Geirhos et al., 2018a). Mummadi et al. (2021), however, did not find a clear correlation between shape bias and robustness.

With respect to adversarial robustness, many shape-biased approaches do yield measurable robustness gains. For instance, Borji (2022) and Sun et al. (2021) showed shape-enhanced models substantially beat vanilla ResNets on FGSM/PGD accuracy, and He et al. (2023a) reported a 15–20 point jump in ImageNet PGD accuracy. These gains are seen in both white-box and transfer-attack settings. On the other hand, several studies (Mummadi et al., 2021; Co et al., 2021; Qiu et al., 2024) caution that shape-bias alone is not a silver bullet: robustness often stems from the training augmentation (stylization) or a blend of features. Naseer et al. (2021) noted that ViTs – which are inherently more shape-biased than CNNs – are much more robust than CNNs under FGSM/PGD attacks, but they also observed a trade-off: training on Stylized ImageNet reduced both CNN and ViT adversarial robustness (i.e. increased shape-bias led to worse performance under attack). In practice, the best defenses integrate both shape and texture information (e.g. ensemble/adaptive models (Co et al., 2021; Qiu et al., 2024)).

Despite these seemingly conflicting findings, it is clear that robust shape-based systems are possible (human vision as an example). The fact that some studies report no robustness gains from adding shape bias to existing models does not

imply that shape information is inherently unhelpful for robustness; rather, it may indicate that current architectures are not well suited to effectively exploit shape cues. We believe that shape bias is a contributing factor to robustness, but not a sufficient one in isolation.

## 2.4. Attempts to Increase Shape Bias

Several studies have attempted to increase shape bias in CNNs through data-centric interventions, including stylized training sets (Geirhos et al., 2018a) (Fig. 11), strong data augmentation (Hermann & Lampinen, 2020), and shape-guided augmentation strategies (Li et al., 2020; Tripathi et al., 2023). Please see Figs. 14 and 15 in Appendix. While these approaches often increase shape bias as measured by cue-conflict metrics, the resulting models frequently remain insensitive to global shape disruptions and fail to exhibit human-like perceptual grouping. Architectural alternatives, such as recurrent CNNs, sparse coding constraints (Li et al., 2023), and large-scale vision transformers (Dehghani et al., 2023), show promise in increasing shape sensitivity, particularly when trained at massive scale. However, even these models often rely on texture under conventional training objectives, suggesting that architectural changes alone are insufficient without corresponding inductive biases and evaluation protocols.

## 2.5. Human Shape Perception and Biological Vision

In contrast to CNNs, shape bias in human vision is well documented across neuroscience, psychophysics, and cognitive psychology (Kucker et al., 2019; Landau et al., 1988; Gershkoff-Stowe & Smith, 2004; Navon, 1977). See Fig. 5. Adults can recognize line drawings as quickly as photographs (Biederman & Ju, 1988), and infants recognize novel objects from sparse contour information (Hochberg & Brooks, 1962). Shape bias is also evident in language acquisition, where children generalize object labels primarily based on shape rather than texture or size (Landau et al., 1988). Neuroscientifically, human shape perception relies on both feed-forward extraction of local features and recurrent processes that support contour integration, perceptual grouping, border ownership, and figure–ground segregation (Elder, 2018; Roelfsema & Houtkamp, 2011; Grossberg & Mingolla, 1985; 1987; Craft et al., 2007). These mechanisms are sensitive to the spatial arrangement of parts and global object structure, reflecting principles studied in Gestalt psychology (Wagemans et al., 2012). Such recurrent grouping processes are largely absent from standard feed-forward CNNs, highlighting a key mechanistic gap between artificial and biological vision. The Configural Shape Score (CSS) tests true shape understanding via object–anagram recognition and shows that common shape-bias methods suppress texture without improving global configural reasoning (Doshi et al., 2025).

## 2.6. Local Versus Global Shape Processing

We distinguish **local shape** as contour fragments, edges, and orientation statistics; **global shape** as topological structure, part relations, and invariance under deformation; and **texture** as the spatial statistics of repeated local patterns (e.g. frequency, granularity, and co-occurrence) that characterize surface appearance largely independent of object geometry. A growing body of work emphasizes the distinction between local and global shape processing. CNNs are often sensitive to local contours but fail when global structure is disrupted while local features are preserved, such as through part scrambling (Baker et al., 2018). Humans show the opposite pattern. Recent work on contour integration further demonstrates that most vision models perform near chance when objects are fragmented, whereas humans remain highly accurate (Lonnqvist et al., 2025). Only large-scale models begin to approach human-like performance, suggesting that global shape understanding may emerge only under specific training regimes and inductive constraints.

## 2.7. Methodological Debates: Conflict versus Reliance

In Table 1, we summarize four main experimental approaches that have been used to evaluate shape processing in neural networks. See also Figs. 19, 20, and 21.

Cue-conflict paradigms (Fig. 1.A), in which shape and texture cues are deliberately dissociated (e.g. a cat silhouette with elephant texture), have become a central tool for probing feature reliance in vision models. Across a wide range of studies, CNNs overwhelmingly classify such images according to texture rather than shape (Geirhos et al., 2018a; Baker et al., 2018). Human observers, by contrast, show a strong preference for shape in these same settings.

Although powerful, cue-conflict paradigms are not without limitations. In the Stylized-ImageNet dataset used by Geirhos et al., texture is applied not only to objects but also to the background, introducing additional, potentially unnatural cues for classification. In some cases, the addition of texture suppresses salient edges, resulting in images that are challenging to recognize even for human viewers (Fig. 11 in Appendix). Conversely, cue-suppression approaches (Fig. 1.B) that attempt to remove texture or shape cues (e.g. silhouettes or texture-filtered images) often fail to fully isolate the intended feature, complicating interpretation (Burgert et al., 2025). These concerns motivate the use of simplified or synthetic datasets where shape and texture can be controlled parametrically.

Taken together, prior work establishes that CNNs encode shape information but systematically privilege texture cues at decision time. Apparent contradictions in the literature often arise from differences in experimental paradigms, dataset scale, and the conflation of representation with de-

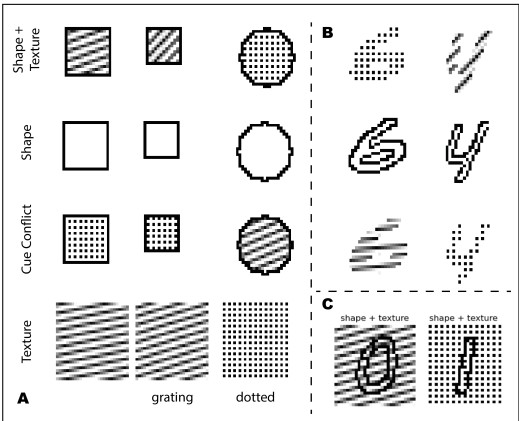

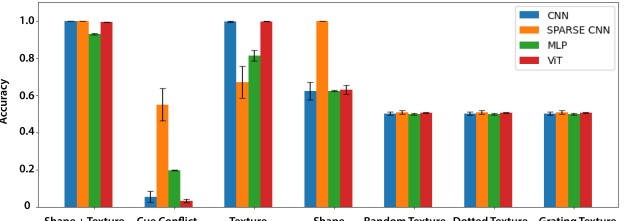

*Figure 2.* A & B) Stimuli used in our case study (grating square vs. dotted circle). Objects appear at varying locations and scales within a 28×28 image. In the texture-only condition, the entire image is textured to avoid inadvertently revealing shape information. Notably, while texture can be removed while preserving shape, the converse is considerably more difficult: eliminating shape while retaining texture is inherently challenging. This asymmetry alone may help explain why human perception is often shape-biased. Our goal here is not to argue that shape is universally the dominant cue, but to test whether CNNs exhibit an inherent structural bias toward shape. A broader, model-independent question concerns whether shape is statistically more informative than texture as a visual cue. C) We also consider a condition in which texture is overlaid on texture. Additional stimuli are provided in Fig. 7 and 9.

cision control. Our work builds on these findings by using controlled stimuli and simplified settings to directly test feature reliance, providing converging evidence that texture bias is a robust and persistent property of CNNs, even when shape information is clearly available.

# 3. Case Study

We introduce minimal, controlled stimuli and evaluate models using both cue-conflict and cue-suppression paradigms. Our goal is not ecological realism but mechanistic isolation: identifying which cues architectures preferentially exploit when confounds are minimized. Unlike prior work, which considers these settings in isolation, we directly compare them and find no discrepancy in the conclusions they yield.

## 3.1. Model Classes

Our primary focus is on CNNs, as convolution constitutes the foundational inductive bias underlying most modern vision systems. CNNs are explicitly designed to exploit local spatial correlations, making them a natural candidate for examining texture-driven behavior.

To contextualize CNN performance, we additionally evaluate vision transformers (ViTs) and multi-layer perceptrons (MLPs) as comparison points (See Appendix). While ViTs rely on self-attention rather than convolution for feature integration, it is important to note that convolution is still commonly used in the patch embedding stage, introduc-

*Figure 3.* Classification accuracy across conditions for CNN, MLP, and ViT models on a binary shape discrimination task (square vs. circle). Error bars denote the standard error of the mean across 10 runs. Results obtained from cue-conflict and cue-suppression paradigms are consistent, showing no discrepancy between the two approaches. Performance in the shape-only condition is low, in agreement with findings from the cue-conflict setting. The SPARSE_CNN exhibits a stronger shape bias, achieving higher accuracy when shape cues are available but at the cost of reduced performance when classification relies on texture cues. Notably, in this task both shape and texture cues are predictive of the class label. See also Fig. 8 and Fig. 9 in Appendix for results on MNIST and FashionMNIST datasets.

ing a degree of local inductive bias. MLP-based models, which lack both convolution and attention-based spatial inductive biases, serve as a further reference point for understanding how architectural assumptions influence cue utilization. In addition, we evaluate a representative model, SPARSE_CNN, which is explicitly designed to promote greater reliance on shape-based cues (Li et al., 2023). By enforcing activation sparsity—typically through masking or thresholding mechanisms—the network limits reliance on dense, texture-like cues and encourages the extraction of global, contour-based features. As a result, the model naturally develops a stronger shape bias while retaining the standard convolutional architecture, demonstrating that architectural inductive biases alone can significantly influence the balance between shape and texture representations. We also consider CapsuleNets (Sabour et al., 2017), whose results are discussed in the Discussion section.

By comparing these model families under identical training and evaluation conditions, our experimental design enables a direct assessment of how architectural inductive biases shape the relative reliance on texture and shape.

## 3.2. Dataset Construction

To isolate and systematically probe the role of shape and texture in visual recognition, we construct a controlled synthetic dataset in which shape and texture cues can be independently manipulated (Fig. 2). Our dataset is inspired by the Navon paradigm, originally proposed by Navon (1977) to investigate how humans process global and local visual information. Please see Fig. 16.

Each image contains a single geometric object belonging to one of two shape classes: a "circle" or a "square". Objects are rendered at varying sizes and placed at random spatial locations within the image to prevent trivial position- or

scale-based solutions. The dataset comprises 5,000 training images and 2,000 test images, with balanced representation across shape and texture classes. All images have a fixed resolution of $28 \times 28$ pixels.

Each shape is consistently paired with a distinct texture during training. Specifically, squares are rendered with a grating texture, while circles are rendered with a dotted texture. This deterministic pairing induces a strong correlation between shape and texture, enabling the model to achieve high training accuracy using either cue. Crucially, this design allows us to later disentangle whether models rely on shape, texture, or a combination of both when these cues are selectively manipulated at test time.

### 3.3. Training Protocol

All models are trained from scratch using the same optimization procedure to ensure comparability across architectures. We use cross-entropy loss for classification and optimize all models with the Adam optimizer using a learning rate of $10^{-3}$. Training is performed for 10 epochs, and each experimental condition is repeated across 10 independent runs with different random initializations to account for optimization variability. We report the mean performance over runs. A batch size of 256 is used throughout. No regularization techniques or data augmentations are employed, allowing us to focus exclusively on the interaction between architectural bias and cue utilization.

### 3.4. Evaluation Scenarios

Models are trained exclusively on images containing both shape and texture cues in their correlated form. After training, we evaluate each model under four distinct test conditions designed to disentangle reliance on shape and texture: 1) **Shape + Texture**: Test images follow the same shape–texture pairing as the training data. This condition measures standard in-distribution performance., 2) **Cue Conflict**: Shape and texture associations are reversed (i.e., circles rendered with grating texture and squares with dotted texture). This condition directly probes which cue dominates the model's decision when shape and texture provide contradictory information, 3) **Texture Only**: Texture cues are preserved, while shape information is removed or rendered uninformative. This condition tests whether texture alone is sufficient for classification. To eliminate explicit shape cues, texture is applied uniformly across the entire image, and 4) **Shape Only**: Shape is preserved while texture cues are removed or neutralized, isolating the model's ability to rely on global shape.

In addition to these core conditions, we include three baseline test sets to contextualize model performance: a) **Random Texture**: Shapes are rendered with randomly assigned textures, breaking any consistent shape–texture association,

b) **Dotted Texture Only**: All shapes are rendered using the dotted texture, removing texture as a discriminative cue, and c) **Grating Texture Only**: All shapes are rendered using the grating texture, similarly eliminating texture-based discrimination. See Fig. 7 in Appendix.

Together, these evaluation scenarios provide a comprehensive picture of how different architectures encode and utilize shape and texture cues under controlled conditions.

### 3.5. Results and Analysis

Results are summarized in Fig. 3. We first present results for CNN, MLP, and ViT models, followed by SPARSE_CNN. Models achieve near-ceiling performance in the shape + texture condition, indicating that all model classes can successfully solve the task when shape and texture cues are aligned. This result is expected, as both cues are fully predictive of the class label in this setting.

In contrast, performance drops sharply in the cue-conflict condition across all standard architectures. Importantly, in this setting global shape remains fully predictive of the correct label for humans, while texture cues are misleading. The substantial reduction in accuracy therefore indicates that models fail to exploit shape information when it competes with texture. Instead, predictions are dominated by texture cues, consistent with prior findings reported by Geirhos et al. (2018a) and Baker et al. (2018). This result provides direct behavioral evidence that, although shape information may be present in the representations, it does not reliably control model decisions under cue competition.

Further support for this interpretation is provided by the texture-only and shape-only evaluation conditions (cue-suppression). Models consistently achieve higher accuracy in the texture-only condition than in the shape-only condition, reinforcing the conclusion that texture cues are more readily exploited than shape cues. Crucially, this pattern aligns with the cue-conflict results and demonstrates that there is no contradiction between cue-conflict and cue-suppression paradigms. While cue-suppression settings—such as those employed in prior work (e.g. Burgert et al. (2025))—have been used to argue for shape bias in CNNs, our results show that shape sensitivity in isolation does not translate into shape dominance when competing cues are present. As shown in Fig 1, the paradigm introduced by Burgert et al. (2025) does not completely remove the shape or texture information, which limits its interpretability; our stimuli, by design, enforce a stricter separation of these cues.

In the random texture, dotted-only, and grating-only baseline conditions, all models perform at approximately chance level, as expected. In these settings, texture cues are either non-informative or uniformly distributed across classes, and

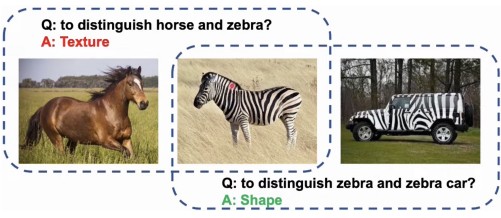

*Figure 4.* Which feature contributes most to distinguish these classes? The answer is likely task dependent.

shape alone is insufficiently exploited by most architectures to support reliable classification.

Viewed differently, our results indicate that both texture and shape contribute to recognition performance. Models achieve their highest accuracy when both cues are present; however, texture is more predictive than shape for the models, whereas the opposite holds for humans.

**Architectural Differences.** Unlike CNNs and ViTs, the MLP relies less on texture, as indicated by its improved performance under cue-conflict conditions and its reduced accuracy in texture-only settings. However, it exhibits lower overall performance when both shape and texture cues are available. Among all evaluated models, the SPARSE_CNN stands out as an effective approach. This model demonstrates substantially higher accuracy in both the cue-conflict and shape-only conditions, alongside reduced reliance on texture cues. These results suggest that sparsity induces a bias toward more global and structured representations, enabling shape information to play a more decisive role in classification. Notably, this shift is achieved without sacrificing in-distribution performance.

**Additional results.** We replicated our experiments on the MNIST and FashionMNIST datasets by applying the same texturization procedure used for our shape dataset. Quantitative results are reported in Fig. 8. Representative examples are shown in Fig. 2(panel B) and Fig. 9. For these datasets, we consider all pairwise class combinations, train a binary classifier for each pair, and average performance across all classifiers. Overall, the qualitative and quantitative trends closely mirror those observed in the shape experiments, providing further evidence that the evaluated models exhibit a systematic bias toward texture cues rather than shape.

Notably, in contrast to earlier settings, the SPARSE_CNN model did not yield performance improvements on these datasets. It is a proof-of-principle that architectural bias matters but it is not yet a general solution. Further, it is not clear whether sparsity truly encourages global structure, or simply suppresses texture statistics. We emphasize that SPARSE_CNN is not offered as a solution to texture bias, but as a lower bound on what architectural modification alone can achieve. Its value is conceptual: it demonstrates that a purely structural change — with no modification to the data, loss function, or training procedure — is sufficient to shift cue reliance in a measurable way. Its failure

to generalize to MNIST and FashionMNIST is itself informative. It confirms that suppressing local activations is not equivalent to building global shape representations, and that closing the gap with human-like shape processing will require more principled architectural innovations — specifically, mechanisms that support spatial integration across extended regions, part-whole relationship encoding, and recurrent feedback. In this sense, SPARSE_CNN sharpens rather than resolves the challenge: it shows the door can be opened by architectural means, while making clear that a more deliberate key is needed to walk through it.

In addition, we considered an alternative experimental condition in which explicit object boundaries were overlaid on texture patterns, as illustrated in Fig. 2 (panel C). This condition more closely mirrors the cue-conflict paradigm. Consistent with our earlier findings, models continued to rely more heavily on texture information than on shape cues. When evaluated using boundary alone, performance dropped to near-chance, indicating that explicit shape cues were insufficient to drive reliable classification (Fig. 10).

## 4. Alternative Views

Two distinct questions are often conflated in the literature. The first asks whether CNNs are inherently shape-biased or texture-biased, while the second concerns which cue is more statistically prevalent and informative in natural images from an information-theoretic perspective. These questions are logically independent, yet frequently treated as equivalent. From an ecological viewpoint, shape may constitute a particularly stable and informative cue, which could explain why human vision relies strongly on it.

Existing work largely reflects two opposing views, arguing either for texture-based or shape-based processing in CNNs. A third perspective, however, emphasizes that the relative importance of shape and texture is task dependent (e.g. Ge et al. (2022)), as illustrated in Fig. 4. Strong texture bias can benefit in-distribution tasks such as fine-grained recognition (e.g. bird classification), whereas shape is critical for tasks requiring robustness and generalization. Accordingly, recent work advocates for balanced cue utilization rather than strict shape dominance.

Hermann & Lampinen (2020) and Hermann et al. (2023) analyze how deep networks rely on different feature sets when performing classification, and argue that this reliance can manifest as so-called shortcut learning. They argue that models tend to rely on features that are easier to extract and optimize, rather than on more complex features that may support stronger generalization. While we broadly agree with this perspective, we do not fully endorse the term shortcut. Models are not "cheating"; instead, they behave exactly as dictated by their architecture, training objective, and gradient-based optimization. From this point of view,

the features they exploit reflect the inductive biases of the model rather than the pathological behavior. Consequently, we argue that simply increasing the amount of data, applying stronger data augmentations, or adding training heuristics is unlikely to overcome the fundamental inductive biases of existing architectures.

Data-driven approaches such as AugMax (Hendrycks et al., 2019) and EquiMod achieve measurable reductions in texture sensitivity without architectural changes. However, these methods induce behavioral invariance without addressing the mechanistic cause — the local receptive field and lack of relational pooling remain intact, leaving models susceptible to re-biasing under distribution shift.

## 5. Call to Action

The preceding analysis identifies texture bias as a fundamental architectural property of CNNs rather than a data artifact. We now translate this diagnosis into a concrete research agenda, organized around three architectural directions, a measurable definition of progress, and two training interventions. See also Appendix 7.2.

**Architectural directions.** We identify three architectural families that directly target the mechanistic deficiencies of standard CNNs identified above:

1. **Equivariant networks.** Group-equivariant convolutional networks (Cohen & Welling, 2016) encode geometric symmetries directly into the architecture, structurally enforcing invariances that standard CNNs must learn statistically from data. These architectures directly address the local receptive field pathology and are expected to yield higher shape integration indices $\Phi(f)$ at equivalent parameter counts.

2. **Part-based and capsule networks.** CapsuleNets (Sabour et al., 2017; Hinton, 2023) explicitly encode part–whole spatial relationships via dynamic routing, preserving the precise geometric structure that pooling operations discard. These architectures are the natural candidate for improving configural shape understanding as measured by CSS (Doshi et al., 2025).

3. **Recurrent and feedback-augmented architectures.** Horizontal and feedback connections, which are largely absent from feedforward CNNs, are critical for contour integration and perceptual grouping in biological vision (Roelfsema & Houtkamp, 2011; Kreiman & Serre, 2020). Architectures that incorporate lateral recurrence (Serre, 2019) are the most promising path toward human-level performance on contour integration benchmarks (Lonnqvist et al., 2025).

**A measurable definition of progress.** We propose that progress be evaluated against a three-level **shape processing ladder**:

*Table 2.* Architectural directions, target metrics on the shape processing ladder (1), and mechanistic deficiencies addressed.

| Direction | Target Metric | Deficiency Addressed |
|---|---|---|
| Equivariant networks | $\Phi(f)\uparrow, \beta_{\text{shape}}\uparrow$ | Local receptive field, pooling |
| CapsuleNets | CSS$\uparrow$ | Part–whole spatial relations |
| Recurrent / feedback | CIS$\uparrow$ | Contour integration, grouping |
| Shape-contrastive loss | $\beta_{\text{shape}}\uparrow$, CSS$\uparrow$ | Training objective misalignment |
| Cue curriculum | All levels$\uparrow$ | Texture shortcut entrenchment |

$$\underbrace{\beta_{\text{shape}}\uparrow}_{\text{Level 1: cue preference}} \longrightarrow \underbrace{\text{CSS}\uparrow}_{\text{Level 2: configural understanding}} \longrightarrow \underbrace{\text{CIS}\uparrow}_{\text{Level 3: contour integration}}$$

(1)

Current architectures, including large-scale ViTs, partially pass Level 1 but fail Levels 2 and 3. We propose that an architecture claiming human-like shape processing must pass all three levels. This gives the community a clear, falsifiable benchmark.

**Training interventions.** Architectural changes must be paired with training objectives that reward global shape reliance. We propose two interventions:

1. **Shape-contrastive loss.** A contrastive term added to the standard cross-entropy loss penalizes representations that are more similar across shape-matched, texture-mismatched pairs than texture-matched, shape-mismatched pairs:

$$\mathcal{L}_{\text{total}} = \mathcal{L}_{\text{CE}} + \lambda \cdot \mathcal{L}_{\text{shape-contrast}}, \qquad (2)$$

where $\mathcal{L}_{\text{shape-contrast}}$ is a margin loss over cue-conflict triplets, operationalizing shape dominance as a training objective rather than an evaluation criterion.

2. **Cue availability curriculum.** Inspired by developmental accounts of shape bias acquisition (Landau et al., 1988), a training curriculum that progressively reduces texture informativeness encourages models to develop shape representations before texture shortcuts become entrenched, connecting to stylized training (Geirhos et al., 2018a) but framing it as a principled schedule rather than a one-shot augmentation.

Table 2 summarizes the agenda, pairing each direction with its target metric and the mechanistic deficiency it addresses.

## 6. Discussion and Conclusion

The investigation into representational biases reveals a fundamental tension at the heart of modern computer vision. As evidenced by the preceding analysis, the debate regarding the visual strategies of DNNs remains unresolved; however, the preponderance of evidence suggests that current architectures are predominantly **texture-biased** (Geirhos et al., 2018a;b). While these models achieve human-level or even super-human performance on closed-set benchmarks, their

reliance on local image statistics rather than global structural integrity exposes a profound divergence from human perception. We argue that the community's current reliance on incremental "patches"—such as data augmentation or auxiliary losses—is insufficient. To achieve truly robust, shape-aware vision, we must move toward fundamental architectural innovations that prioritize global integration over local shortcut exploitation.

**The Problem of Definition: Texture, Shape, and Entanglement.** A core source of confusion in the literature stems from inconsistent definitions of "shape" and "texture." See section 2.6 for definitions. Unlike texture, shape is inherently long-range; it requires the integration of spatial information that may be separated by large distances in the image plane. In natural images, these two properties are often entangled, making it difficult to isolate the true driver of a model's prediction. This entanglement often leads to misinterpreted results: success on sketches or silhouette-based tasks does not necessarily imply a genuine understanding of shape. Models may still exploit unintended cues, such as interior statistics or background regions, leaving them vulnerable to adversarial manipulation that preserves shape but alters local patterns.

**The Limits of Data-Driven Invariance.** It is common to rely on data augmentation when principled mechanisms for inducing invariance are lacking; for example, rotation or scale invariance is encouraged by augmenting datasets with transformed images. This "brute-force" strategy contrasts with the human visual system, where invariance is a structural property of perception: neurons in the visual cortex respond across multiple scales, orientations, and locations. Although techniques such as random erasing, adversarial training, and stylization can encourage shape reliance by suppressing texture shortcuts, they act mainly as regularizers and do not change the underlying convolutional computation. Consequently, the lack of robust shape representations remains a critical vulnerability, especially in Out-of-Distribution (OOD) and safety-critical settings such as autonomous driving, where brittle texture cues can lead to catastrophic errors.

We believe that approaches that explicitly exploit geometry and symmetry, such as group convolutions (Cohen & Welling, 2016) or CapsuleNets (Sabour et al., 2017; Hinton, 2023) (See Appx. 7.1 for results using this architecture), constitute a promising direction for improving both the efficiency and generalization of learning systems. By encoding invariances and equivariances directly into the model architecture, these methods reduce sample complexity, improve robustness, and align inductive biases more closely with the underlying structure of the data.

**Architectural Origins of Bias.** Texture bias is not merely a data problem; it is rooted in the inductive biases of convolutional architectures. Local receptive fields prioritize fine-grained statistics, while pooling operations—designed to provide translation invariance—often discard the precise spatial relationships essential for global shape (Baker et al., 2020). This is consistent with classic findings showing that CNNs struggle with relational tasks and spatial reasoning (Fleuret et al., 2011). Even the shift toward ViTs provides only a partial solution. While self-attention allows for holistic integration, ViTs remain surprisingly insensitive to certain patch-based transformations that destroy semantics for human observers (Qin et al., 2022) (Fig. 18). This suggests that even non-convolutional models can fall into the trap of exploiting local, non-robust features if the training objective does not explicitly penalize them. We should move beyond merely measuring bias toward understanding and controlling the learning dynamics that drive feature reliance, using richer metrics that capture multidimensional feature use (e.g. occlusion, parts, affordances) and studying how dataset scale interacts with architectural inductive biases to enable more human-like representations.

**Biological Inspiration: Missing Mechanisms.** Despite their impressive capabilities, CNNs and Transformers struggle with cluttered scenes (Volokitin et al., 2017), out-of-context objects (Rosenfeld et al., 2018), and minimal recognizable images (Ullman et al., 2016). Evidence from visual neuroscience highlights several critical components missing from deep architectures that contribute to human-like robustness: 1) **Contour Integration:** Human vision remains accurate even when objects are fragmented or contours are sparse (Fig. 12). Most vision models perform near chance in these conditions, failing to integrate fragments into a global whole (Lonnqvist et al., 2025), 2) **Feedback and Recurrence:** The visual cortex is characterized by extensive horizontal and feedback connections, which are more prevalent than feed-forward ones. These mechanisms are crucial for boundary ownership, figure-ground segregation, and perceptual grouping (Kreiman & Serre, 2020; Serre, 2019), 3) **Task-Driven Attention:** Unlike the passive attention of modern models, human attention is task-driven and sequential, processing objects through gaze and active selection (Itti & Koch, 2001), and 4) **Background Suppression:** Human vision effectively suppresses background noise to focus on foreground structure. Explicitly incorporating this strategy into DNNs has been shown to enhance adversarial robustness (Borji, 2021).

**Bottom Line.** Achieving human-aligned vision requires a shift in humility. We must acknowledge that the problem is still largely open. Future progress demands architectures that prioritize structure over shortcuts and training regimes that value human-like relational reasoning over fragile statistical correlations. Ultimately, solving the shape-texture discrepancy may be the key to unlocking the next generation of robust, trustworthy, and truly intelligent vision systems.

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

# 7. Appendix

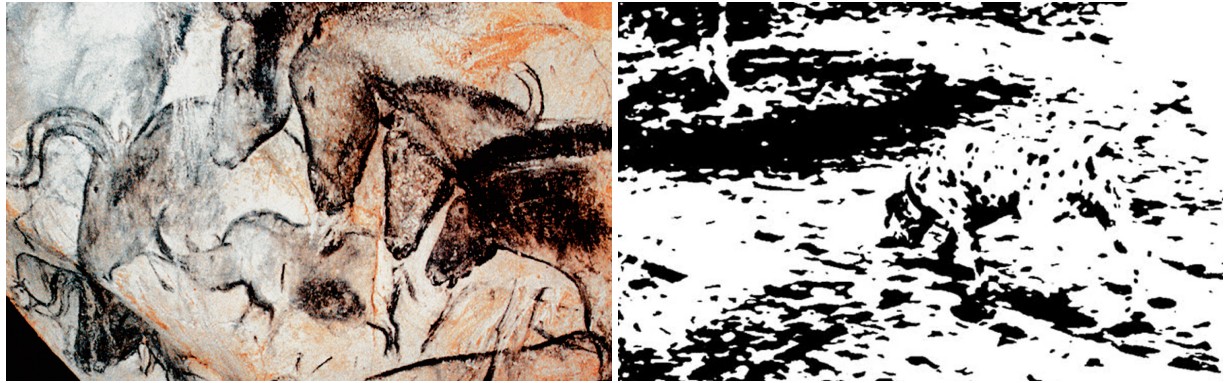

*Figure 5.* Left: Prehistoric cave art from Chauvet Cave (France, ca. 30,000 B.C.). Right: An image of a Dalmatian dog whose distinctive black-and-white texture partially obscures its form, yet its overall shape remains recognizable—head lowered, facing left, and angled slightly toward the viewer. Both images highlight the importance of shape in visual processing.

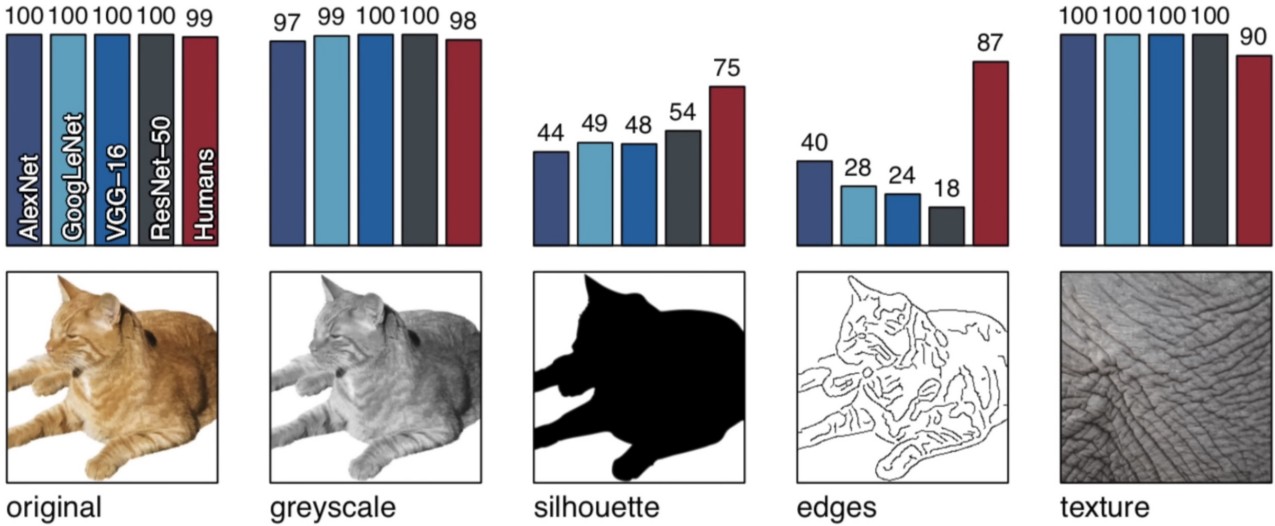

*Figure 6.* Accuracies and example stimuli for five experimental conditions without cue conflict. Figure adapted from Geirhos et al. (2018a).

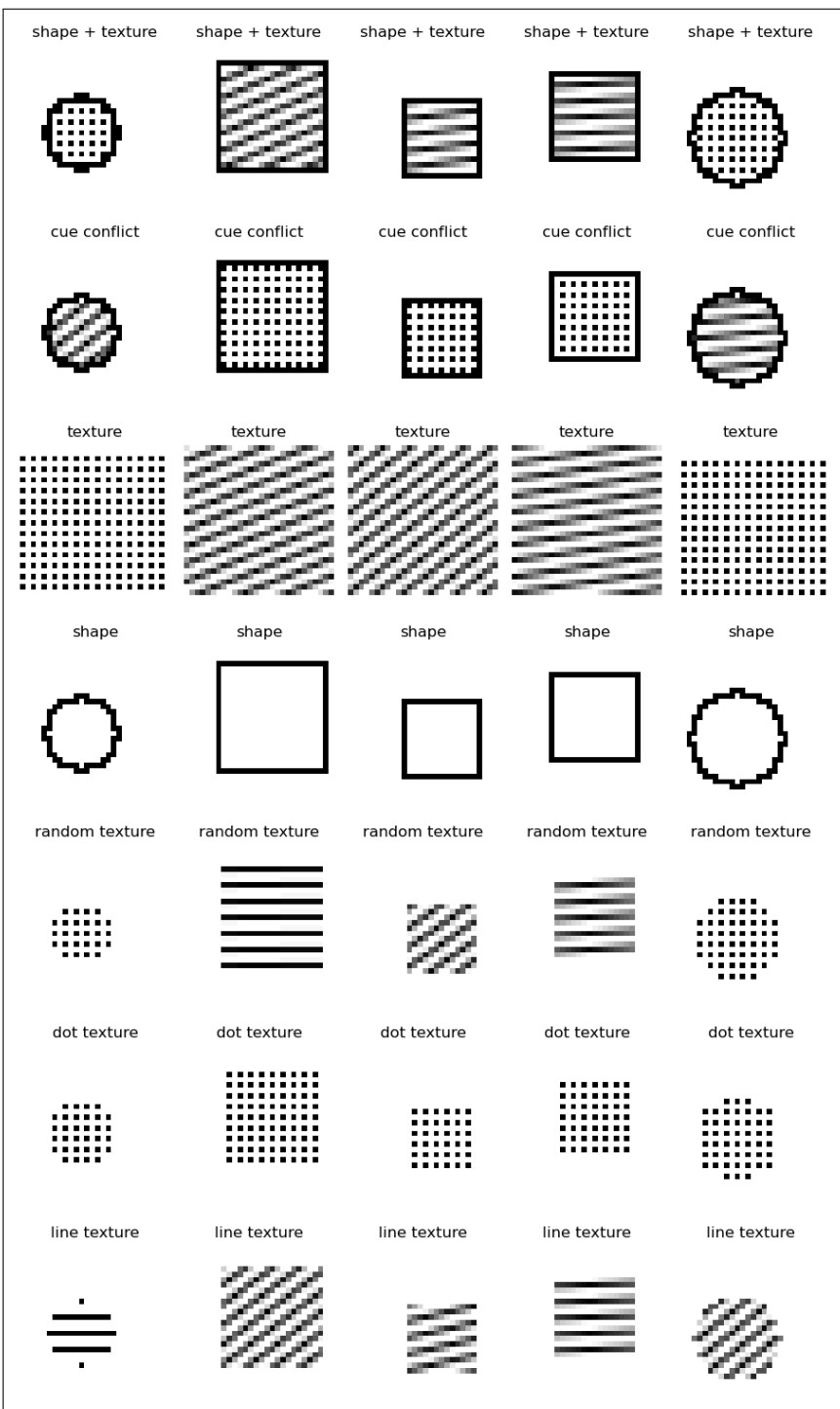

*Figure 7.* Additional stimuli from our dataset for binary shape classification. Line means grating pattern.

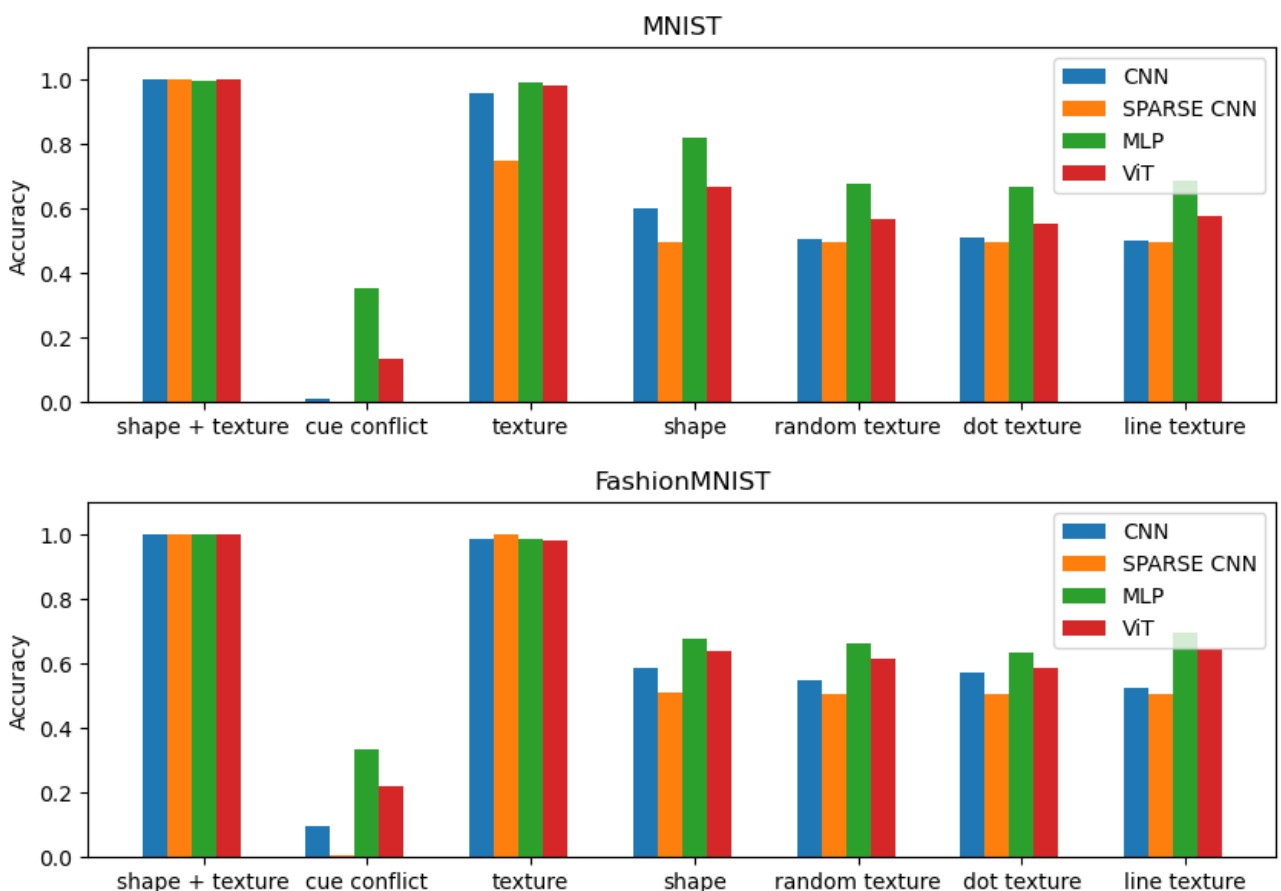

*Figure 8.* Results on MNIST and FashionMNIST datasets. Example stimuli are shown in figures below.

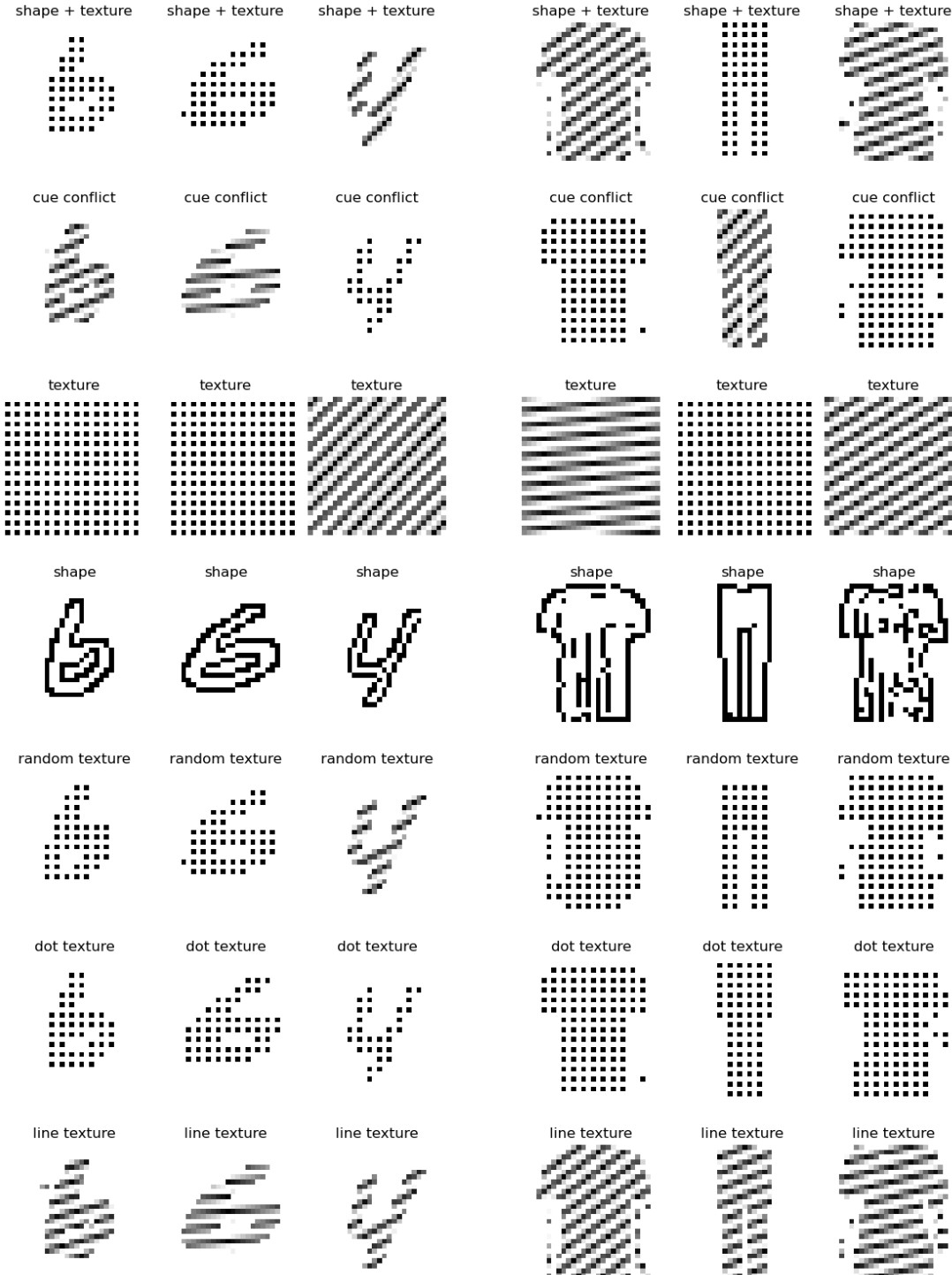

*Figure 9.* Stimuli on MNIST and FashionMNIST datasets. Unlike the shape stimuli, we did not place a black edge around the digit, as doing so would eliminate much of its internal texture — primarily because digits are small. Nonetheless, shape information remains available from the implicit contour surrounding the textured digit.

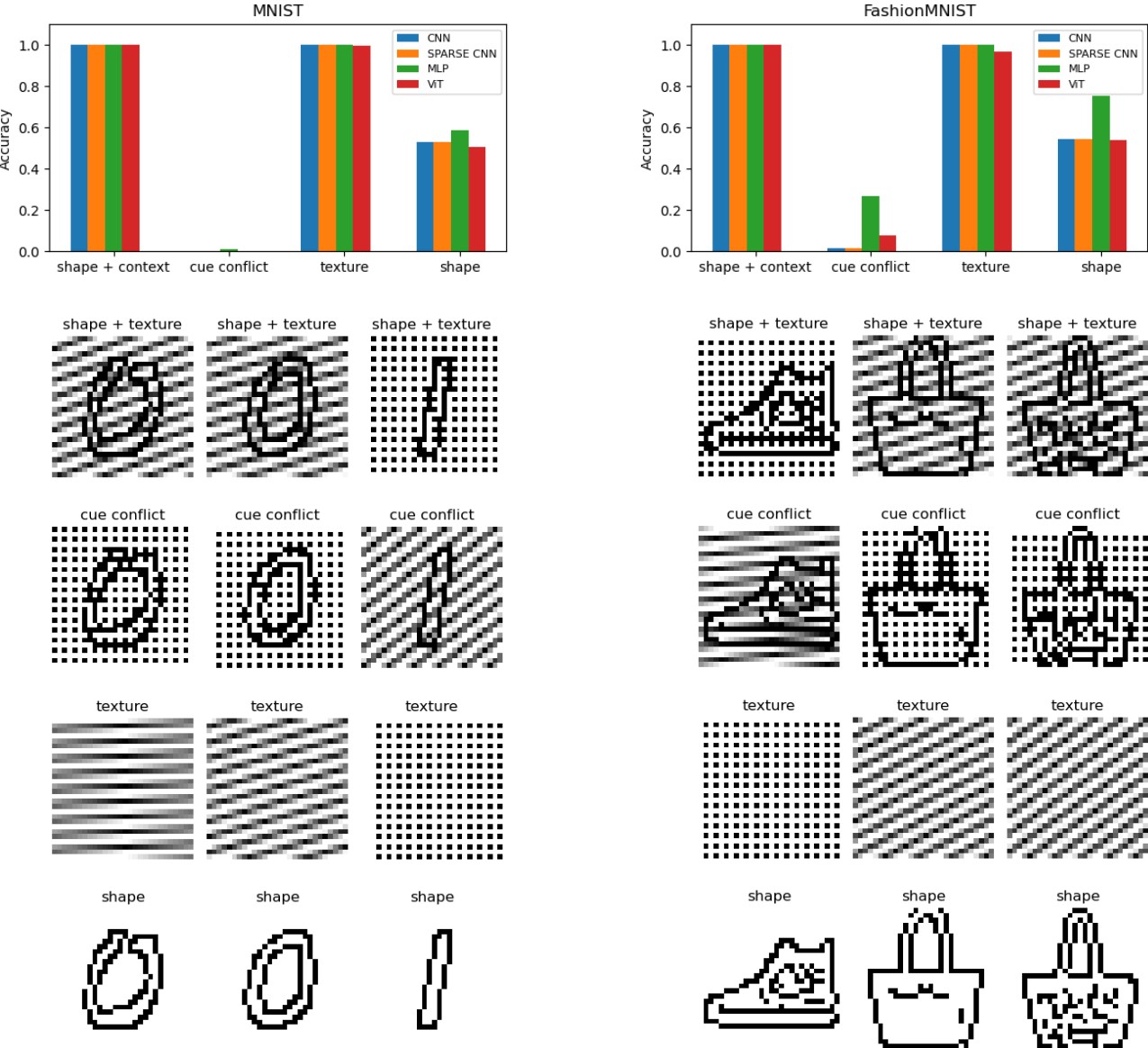

*Figure 10.* Results and stimuli for a variation of MNIST and FashionMNIST in which shape is overlaid on texture. Binary models are trained on all pairs of classes.

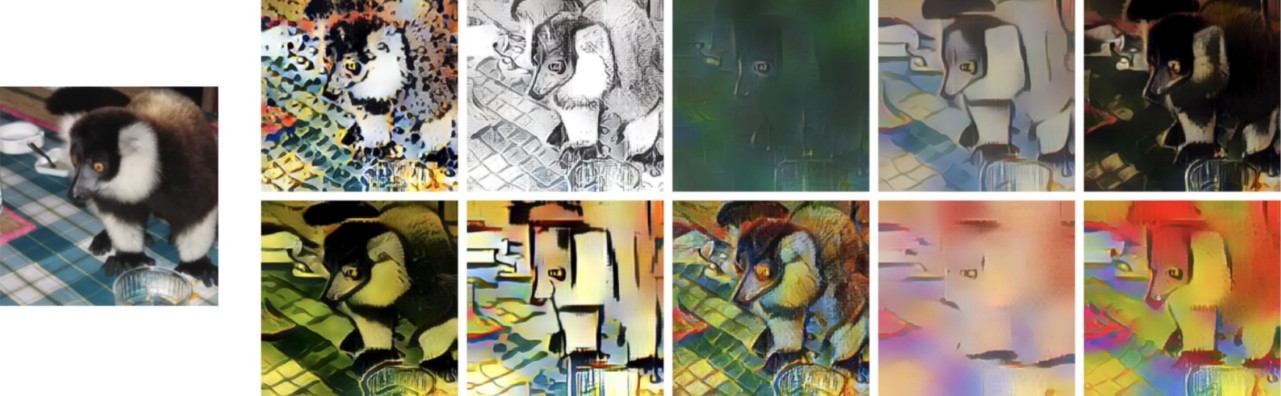

*Figure 11.* Stylized ImageNet (SIN) examples produced using AdaIN style transfer. Stylization replaces local texture cues while preserving global object shape, thereby reducing texture predictiveness for classification. Each ImageNet image is stylized once, using a single randomly selected painting style. Figure from Geirhos et al. (2018a).

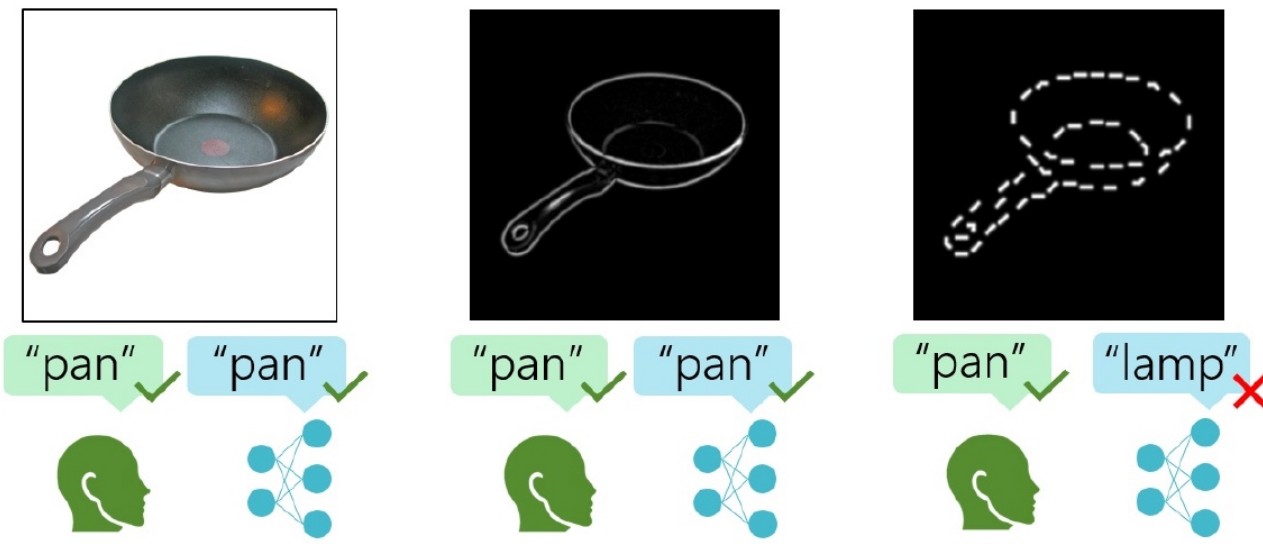

*Figure 12.* Comparison of human and DNN categorization for **(Left)** RGB images, **(Middle)** contour-extracted images, and **(Right)** fragmented images that require contour integration. Over 1,000 tested models fail dramatically once object contours are disrupted. Figure from Lonnqvist et al. (2025).

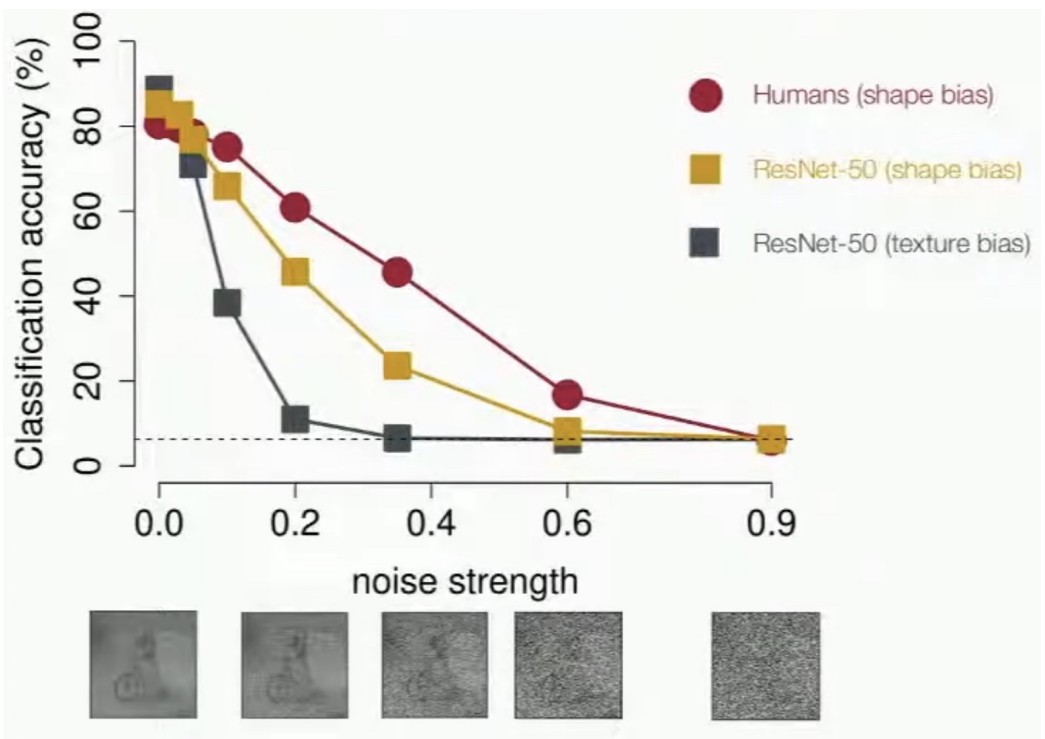

*Figure 13.* Sensitivity of humans and models (original and shape-biased) to noise. Figure from Geirhos et al. (2018b).

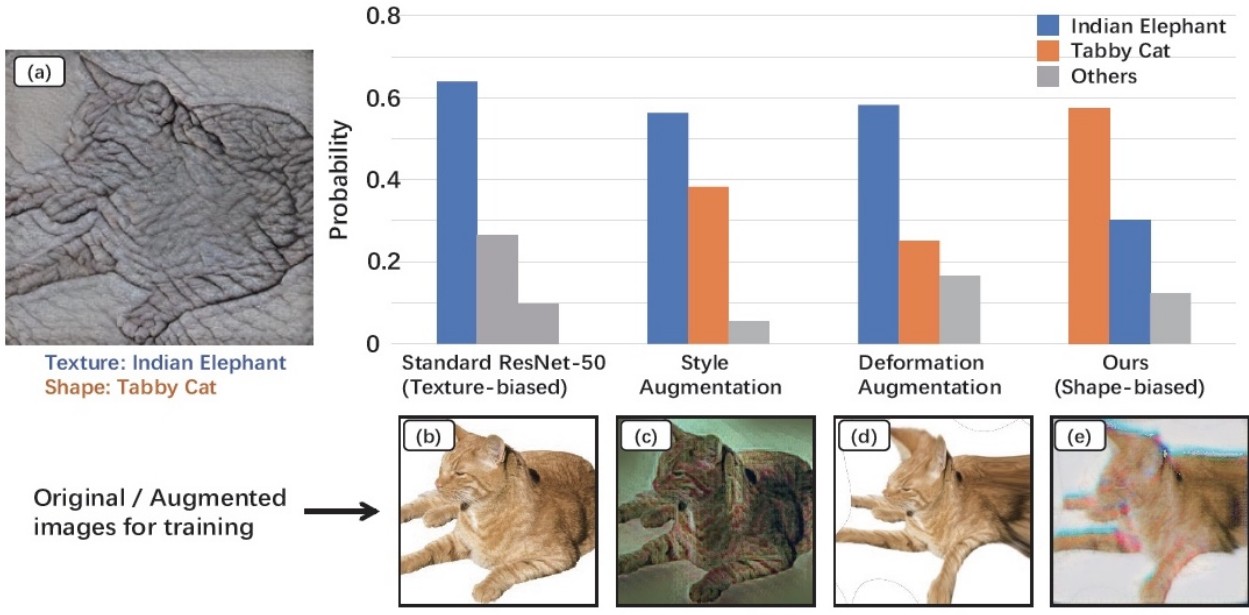

*Figure 14.* A shape-based augmentation method to enhance model shape-bias. Figure from He et al. (2023b).

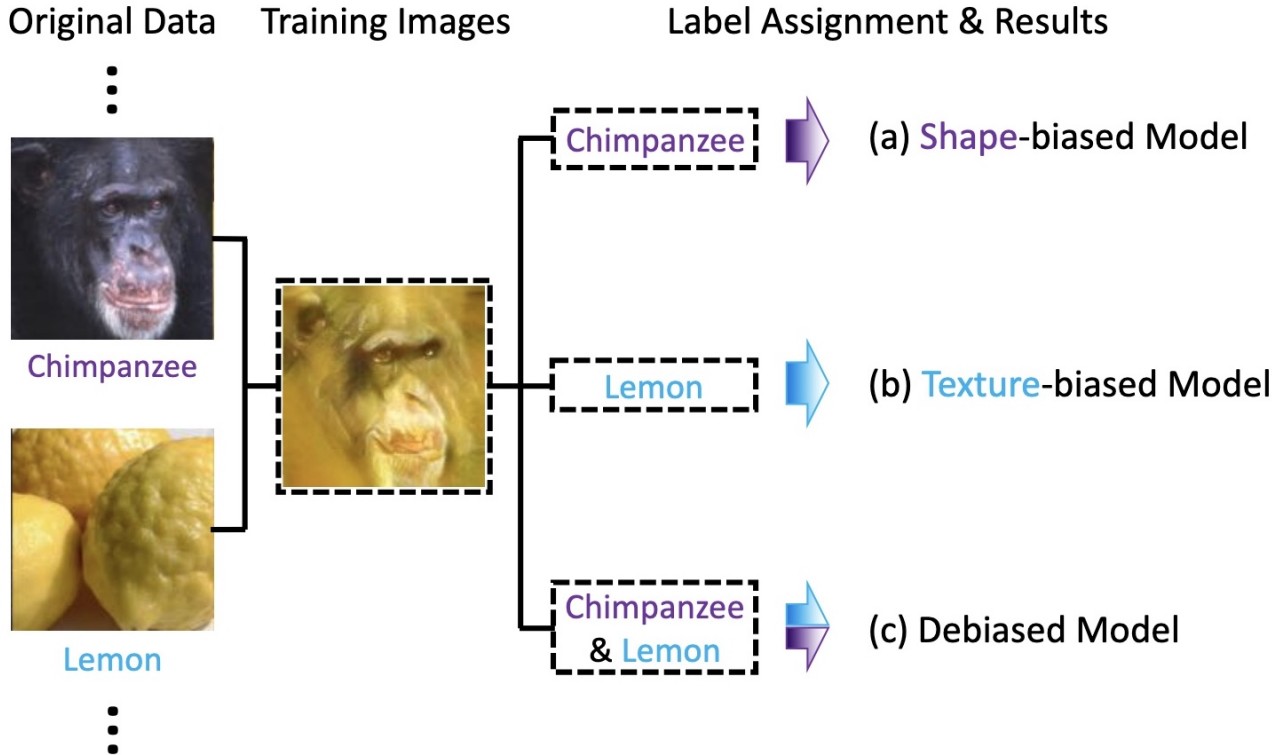

*Figure 15.* Illustration of the training pipeline of Li et al. (2020) for acquiring (a) a shape-biased model, (b) a texture-biased model, and (c) a shape-texture debiased model. All models are trained on the same set of images with conflicting shape and texture information, generated via style transfer between two randomly selected images. They differ in their labeling strategies: for (a) and (b), labels are assigned according to the image providing shape or texture information, guiding the models to emphasize shape or texture features; for (c), labels are determined jointly from both images, encouraging the model to avoid bias in representation learning.

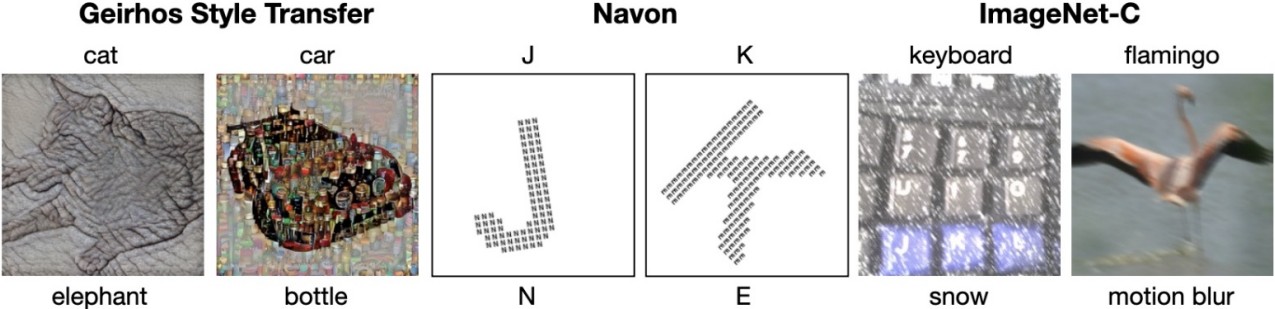

*Figure 16.* Sample items from the three datasets, labeled by shape (top) and texture (bottom). A Navon figure consists of a large, globally recognizable shape (e.g. a letter) constructed from repeated instances of a smaller, different shape (https://en.wikipedia.org/wiki/Navon_figure). Figure from Hermann & Lampinen (2020).

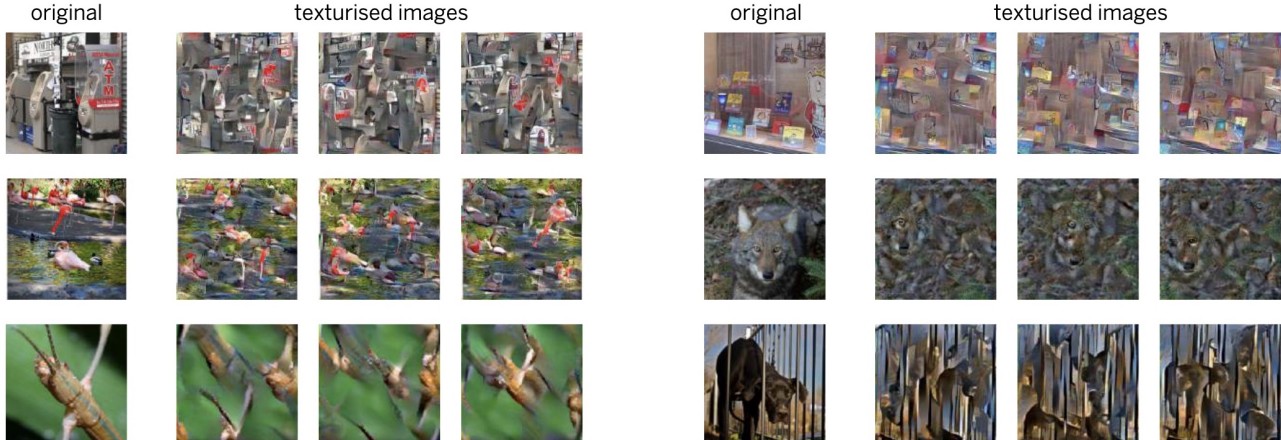

*Figure 17.* Examples of original and texturized images. A standard VGG-16 maintains high accuracy on the texturized images, whereas humans' performance drops significantly due to the loss of global shape information in many cases. Figure from Brendel & Bethge (2019).

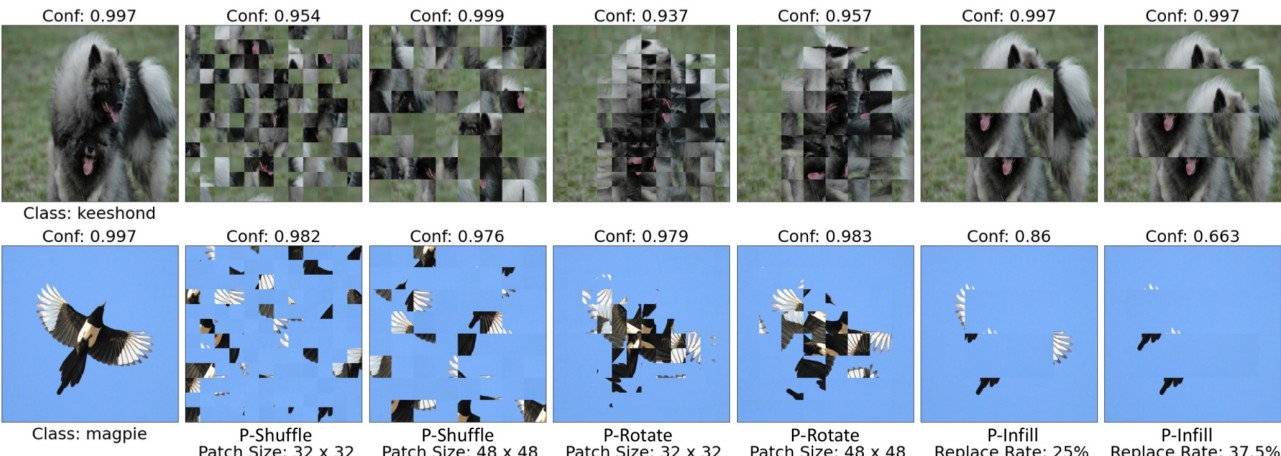

*Figure 18.* Patch-based transformations can render images unrecognizable to humans, yet Vision Transformers (ViTs) still classify them with high confidence (e.g. keeshond or magpie). Visualization of these transformations shows the predicted confidence scores of ViT-B/16 pretrained on ImageNet-21k and finetuned on ImageNet-1k displayed above each image. Qin et al. (2022) investigate ViTs' robustness through their patch-based architecture, which processes images as sequences of patches. Surprisingly, ViTs remain largely insensitive to patch-based transformations, even when the transformations destroy the original semantics and make the image unrecognizable to humans. This suggests that ViTs rely on features that survive such transformations but are generally non-semantic from a human perspective. Further analysis shows these features are useful but non-robust: models trained on them achieve high in-distribution accuracy yet fail under distribution shifts. Their results provide concrete evidence that ViTs depend on non-robust features for predictions, limiting their out-of-distribution robustness.

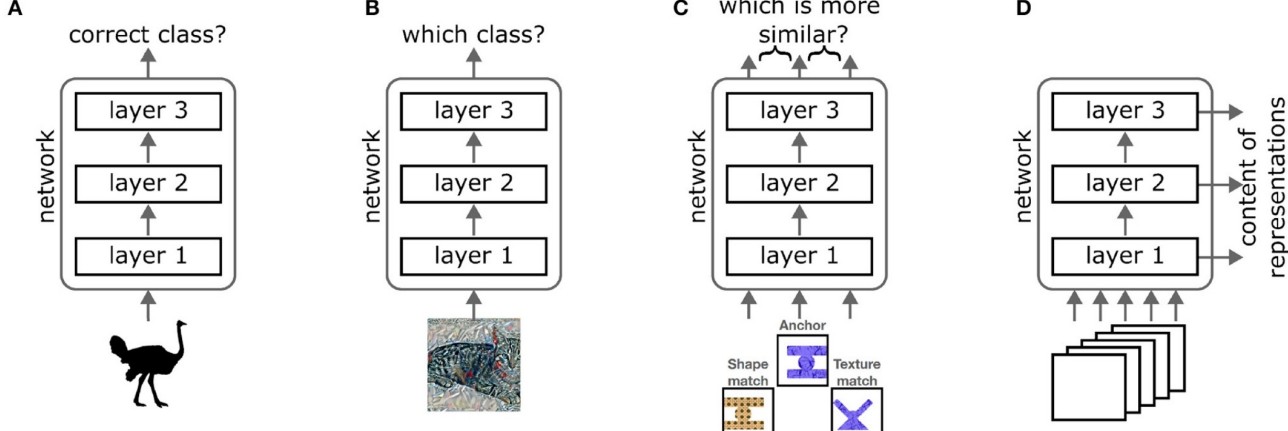

*Figure 19.* Overview of the experimental paradigms used to assess shape processing in neural networks. (A) Diagnostic stimuli limit the information available in an image; for example, silhouettes show only the object's shape. Correct classification of such stimuli indicates that the network can utilize the available cues. (B) Cue conflict stimuli combine features from two different classes, such as the shape of a cat with the texture of a bicycle. The network's choice reveals which feature it prioritizes. (C) Triplet tests present an anchor stimulus alongside two matches that differ from the anchor along distinct feature dimensions. Comparing the network's representations indicates which features it uses to group stimuli. (D) Representation analysis methods record intermediate layer outputs across many images to determine which information is encoded in the network's internal representations. Figure from Jarvers & Neumann (2023).

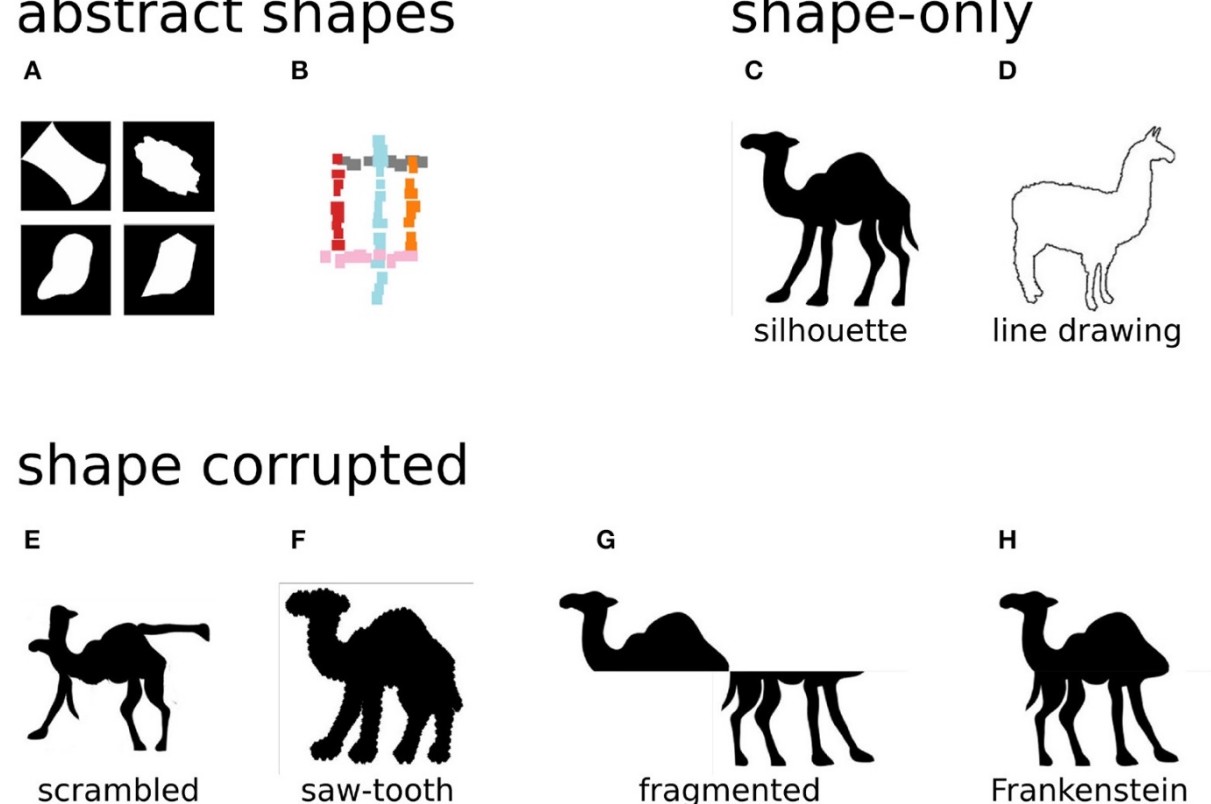

*Figure 20.* Example diagnostic stimuli designed to isolate or manipulate shape information in the literature. Artificial, abstract shapes allow precise control over the shape features present: (A) Kalfas et al. (2018) generated shapes in four categories—regular (top left), complex (top right), simple curved (bottom left), and simple straight (bottom right); (B) Malhotra et al. (2020) created artificial shapes that could be classified based on shape or another feature, such as the presence of a red segment. Shape-only stimuli preserve real-world objects but remove all information except shape, for example, through silhouettes (C) or line drawings (D). Shape-corrupted stimuli further manipulate shape, for instance by scrambling (E) or distorting boundaries (F). Similarly, Baker & Elder (2022) used fragmented silhouettes (G) and re-aligned parts to generate "Frankenstein" stimuli (H). Figure from Jarvers & Neumann (2023).

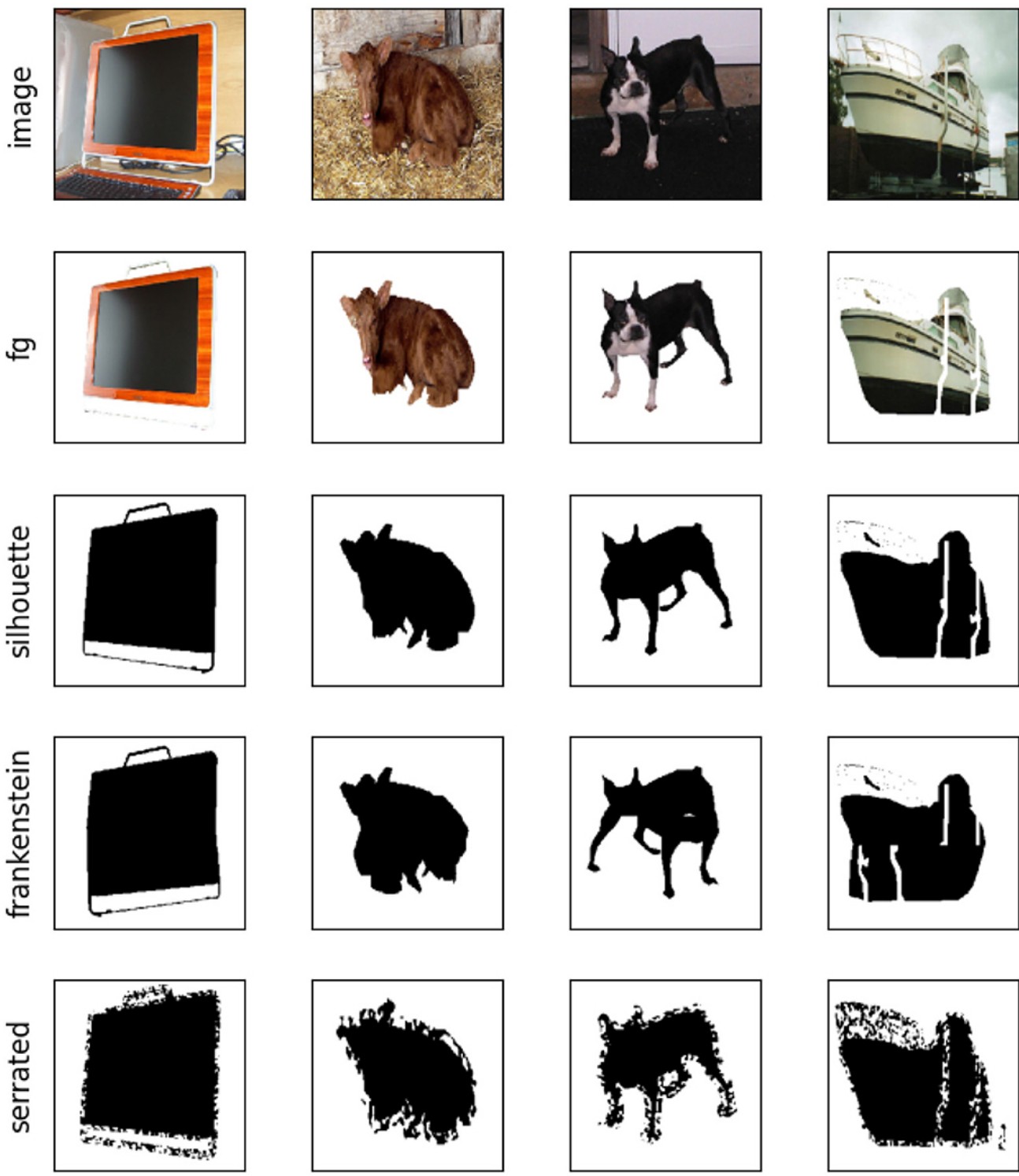

*Figure 21.* Examples of diagnostic stimuli used by Jarvers & Neumann (2023). Each row corresponds to a different stimulus type, with four example images shown in columns. The original image serves as a reference. Foreground images ('fg') mask everything except the object of interest, allowing assessing the influence of background on responses and representations. Silhouettes fill all object pixels with black, preserving only shape information. Frankenstein stimuli alter the global arrangement of object parts while largely retaining local shape features. In serrated silhouettes, local shape cues are disrupted, but the global shape is preserved.

Python code for models, in order: CNN, VIT, MLP, and SPARSE_CNN (a.k.a, Top_K).

```python
class SimpleCNN(nn.Module):
    def __init__(self):
        super().__init__()
        self.conv = nn.Sequential(
            nn.Conv2d(1, 4, kernel_size=5, padding=2), nn.ReLU(),
# First layer: 4 filters
            nn.MaxPool2d(2),
            nn.Conv2d(4, 8, kernel_size=5, padding=2), nn.ReLU(),
# Second layer: 8 filters
            nn.MaxPool2d(2),
        )
        self.fc = nn.Sequential(
            nn.Flatten(),
            nn.Linear(8 * 7 * 7, 32), nn.ReLU(),  # Adjusted for final feature map size
            nn.Linear(32, 2)  # Output for 2 classes
        )

    def forward(self, x):
        return self.fc(self.conv(x))

class SimpleViT(nn.Module):
    def __init__(self, img_size=28, patch_size=7, dim=64, depth=4, heads=4, mlp_dim=128, num_classes=2):
        super().__init__()
        assert img_size % patch_size == 0, "Image size must be divisible by patch size"
        num_patches = (img_size // patch_size) ** 2
        patch_dim = patch_size * patch_size  # For grayscale MNIST (1 channel)

        # Patch embedding
        self.patch_embed = nn.Linear(patch_dim, dim)

        # Positional embeddings
        self.pos_embedding = nn.Parameter(torch.randn(1, num_patches + 1, dim))
        self.cls_token = nn.Parameter(torch.randn(1, 1, dim))

        # Transformer encoder
        encoder_layer = nn.TransformerEncoderLayer(d_model=dim, nhead=heads, dim_feedforward=mlp_dim)
        self.transformer = nn.TransformerEncoder(encoder_layer, num_layers=depth)

        # Classification head
        self.fc = nn.Linear(dim, num_classes)

class SimpleMLP(nn.Module):
    def __init__(self):
        super(SimpleMLP, self).__init__()
        self.fc1 = nn.Linear(28 * 28, 128)  # First hidden layer
        self.fc2 = nn.Linear(128, 64)        # Second hidden layer
        self.fc3 = nn.Linear(64, 2)          # Output layer (2 classes)
        self.relu = nn.ReLU()

    def forward(self, x):
        x = x.view(x.size(0), -1)            # Flatten: (batch, 784)
        # print(x.shape)
        x = self.relu(self.fc1(x))
```

```python
        x = self.relu(self.fc2(x))
        x = self.fc3(x)                         # No softmax here (handled by loss)
        return x

class SimpleBlock(nn.Module):
    def __init__(self, in_planes, planes, stride=1):
        super().__init__()
        self.conv1 = nn.Conv2d(in_planes, planes, 3, stride=stride, padding=1, bias=False)
        self.bn1   = nn.BatchNorm2d(planes)
        self.conv2 = nn.Conv2d(planes, planes, 3, padding=1, bias=False)
        self.bn2   = nn.BatchNorm2d(planes)

    def forward(self, x):
        x = F.relu(self.bn1(self.conv1(x)))
        x = F.relu(self.bn2(self.conv2(x)))
        return x

def sparse_hw(x, topk=0.1, tau=1.0, device='cuda'):
    n, c, h, w = x.shape
    if topk == 1:
        return x

    x_ = x.view(n, c, h * w)
    keep = int(max(1, topk * h * w))

    _, idx = torch.topk(x_.abs(), keep, dim=2)
    mask = torch.zeros_like(x_).scatter_(2, idx, 1).to(device)

    sparse = mask * x_

    if tau == 1:
        return sparse.view(n, c, h, w)

    return sparse.view(n, c, h, w) * tau + x * (1 - tau)

class SimpleTopKNet(nn.Module):
    def __init__(self, topk_layers=[1], topk=0.1, num_classes=2):
        super().__init__()

        self.topk_layers = set(topk_layers)
        self.topk = topk

        self.conv1 = nn.Conv2d(1, 32, 3, padding=1, bias=False)
        self.bn1   = nn.BatchNorm2d(32)

        self.layer1 = SimpleBlock(32, 64, stride=1)
        self.layer2 = SimpleBlock(64, 128, stride=2)
        self.layer3 = SimpleBlock(128, 256, stride=2)

        self.fc = nn.Linear(256, num_classes)
```

## 7.1. Performance of CapsuleNet Architecture

Here, we report CapsuleNet results on our synthetic shape dataset under identical training conditions as other models. The results are informative and partially support our hypothesis.

*Table 3.* Classification accuracy (%) across stimulus conditions for CapsuleNet compared to standard CNNs and ViTs.

| Model | Shape+Texture | Texture Only | Shape Only | Cue Conflict |
|---|---|---|---|---|
| Standard CNNs | ∼100.0 | ∼100.0 | ∼chance | ∼10.0 |
| ViTs | ∼100.0 | ∼100.0 | ∼chance | ∼10.0 |
| CapsuleNet | ∼100.0 | ∼100.0 | 91.2 | 12.8 |

CapsuleNet achieves near-perfect performance in the shape+texture condition (100%) and texture-only condition (100%), consistent with all other architectures. The most striking result is its shape-only accuracy of 91.2%, substantially higher than standard CNNs and ViTs, which perform near chance in this condition. This directly demonstrates that CapsuleNet encodes shape information in a way that is actionable for classification — a capability that standard feedforward architectures largely lack.

Under cue conflict, CapsuleNet achieves 12.8%, which while still indicating a degree of texture dominance, is notably higher than the near-zero performance of standard CNNs and ViTs in this condition. This suggests that shape representations in CapsuleNet exert a measurable influence on decisions even under competition with texture, consistent with its stronger part-whole encoding.

CapsuleNet does not fully resolve texture bias — a truly shape-dominant model would approach 100% under cue conflict. However, the gap between CapsuleNet and standard CNNs on both the shape-only and cue-conflict conditions provides concrete empirical support for our central argument: that architectural choices meaningfully determine the degree to which shape information controls model decisions.

We note that some of the figures reported for these models are estimates rather than exact values. We therefore encourage researchers to treat these results as indicative rather than definitive, and to conduct more rigorous and systematic evaluations of these architectures on similar datasets before drawing strong conclusions. Such efforts would help establish more reliable benchmarks and enable fairer comparisons across models.

### 7.2. Mathematical Formalism and Perceptual Metrics

**On mathematical formalism.** The core claim of our paper—that CNNs preferentially exploit texture over shape as a decision cue—can be grounded more rigorously using an information-theoretic framework. Let $X$ denote an input image, $Y$ the class label, $S(X)$ a shape representation (*e.g.*, a silhouette or contour map), and $T(X)$ a texture representation (*e.g.*, a bag-of-local-features or Gram matrix statistic). The texture bias of a model $f$ can be formalized as the degree to which $f$'s predictions are explained by $T(X)$ rather than $S(X)$, measured via the conditional mutual information gap:

$$\Delta I = I\big(f(X); Y \mid S(X)\big) - I\big(f(X); Y \mid T(X)\big). \tag{3}$$

A texture-biased model satisfies $\Delta I \ll 0$: knowing $T(X)$ renders $f$'s prediction nearly redundant, while $S(X)$ alone leaves substantial uncertainty. Human vision, conversely, satisfies $\Delta I \gg 0$. This formalism makes the shape–texture gap precise and measurable, and directly captures what cue-conflict experiments operationalize behaviorally.

Additionally, the architectural origin of texture bias can be formalized through the lens of inductive bias and sufficient statistics. A convolutional architecture with kernel size $k$ and $L$ layers has a theoretical receptive field of $\mathcal{O}(k \cdot L)$, but the *effective* receptive field—the region that causally influences a prediction—is typically far smaller due to the dominance of central weights after pooling (Luo et al., 2016; Le & Borji, 2017). We define a **shape integration index**:

$$\Phi(f) = \mathbb{E}_x \left[ \frac{|\text{eff-RF}(f, x)|}{d(x)^2} \right], \tag{4}$$

where $d(x)$ is the spatial extent of the discriminative object region and $|\text{eff-RF}|$ is the effective receptive field area. For standard CNNs, $\Phi(f) \ll 1$, meaning decisions are driven by regions far smaller than the object—consistent with texture-based processing. A model capable of human-like global shape integration should approach $\Phi(f) \approx 1$. This index would serve as a diagnostic target for architectural interventions and would allow the gap between current architectures and human-level shape integration to be quantified in a hardware- and dataset-agnostic manner.

**On perceptual metrics.** The standard shape bias score $\beta_{\text{shape}}$—defined as the fraction of cue-conflict trials on which a model's prediction agrees with the shape class—is a necessary but insufficient metric. It conflates sensitivity to local contour fragments with genuine global shape understanding, precisely the distinction our paper argues is critical. We propose that a complete evaluation suite should include at minimum three complementary metrics:

1. **Shape Bias Score** $\beta_{\text{shape}}$ (Geirhos et al., 2018a): fraction of cue-conflict classifications consistent with shape. Captures relative preference but not the *type* of shape processing.

2. **Configural Shape Score (CSS)** (Doshi et al., 2025): probes holistic, relational shape understanding via object-anagram recognition, where local shape features are preserved but global spatial arrangement is disrupted. A model relying on contour fragments rather than global structure will fail here despite appearing shape-biased on $\beta_{\text{shape}}$.

3. **Contour Integration Score (CIS)** (Lonnqvist et al., 2025): accuracy on fragmented contour stimuli where global object identity must be inferred by integrating spatially separated fragments. This is the most stringent test of human-like shape processing; over 1,000 tested models perform near chance, while humans remain highly accurate.

Together, these three metrics define a **shape processing ladder**: a model can score highly on $\beta_{\text{shape}}$ while failing CSS (local contour sensitivity without configural understanding), and can pass CSS while failing CIS (configural sensitivity within intact objects, but no contour integration across fragments). Human vision passes all three. We would add a figure in revision illustrating where current architectures—CNNs, ViTs, and SPARSE CNN—sit on this ladder, making the gap explicit and the target for future architectural work concrete.

