# OpenReview forum: "Position: CNNs Don't See Shape — And That Won't Change Without New Architectures"
_ICML.cc/2026/Position_Paper_Track — ICML 2026 Position Paper Track regular_

### Official Review · Reviewer_ZSwF · 2026-03-10

**Significance:** 2
**Argument Clarity:** 2
**Rating:** 3
**Confidence:** 4

**Questions:**

Please see the above comments.

**Alternative Views Section:**

Yes

**Compliance With Llm Reviewing Policy A Conservative:**

Affirmed.

**Discussion Potential:**

2

**Paper Summary:**

This paper investigates the widely studied problem, if shape or texture cues are dominant in a learned model.  This draft uses several pages to revisit or survey the previous shape-texture studies. Then a new shape-texture dataset, grating square vs. dotted circle, is proposed to test the visual cues. And this study shows that “CNNs consistently prioritize texture over global shape when cues compete, even when shape information is explicitly available.”

**Position:**

No

**Position In Title:**

No

**Related Work:**

3

**Strengths And Weaknesses:**

This draft attempts to study a very important concept in computer vision, shape vs texture in a learned model. However, a main problem in this draft is that the "position" of the study is not clear.

From the title, "Evidence and Implications of Texture Bias in Deep Neural Networks". Does this study echo the concept that texture bias is presented in networks? or this study finds new proof to show case the texture bias? Either way, why is this a new position? From the abstract, "We argue that much of this apparent discrepancy reflects methodological confounds...", throughout section 2, different types of methodologies are listed and surveyed, the main points are recapped in the their OLD POSITIONS. All of the claims, strengths or limitations are already shown and discussed in the previous studies, it is not clear what the new position is in this paper.

Line 97, "We critically review prior experimental paradigms, outlining their respective strengths and limitations, and then demonstrate...", the critical reveiws seem to be a summary of the previous studies, the limitation listed is already known. Especially, line 102, "We take the position that purely feed-forward convolutional networks..." and line 108, "We argue that resolving the shape–texture debate requires controlled experiments...", they are widely studied in the previous studies.

The only thing new proposed in this study is the grating squre vs. dotted circle (Fig 2) to support the claim "texture bias is a natural consequence of the local inductive bias of convolution." Please correct me if my understanding is wrong, the shape cue is square vs circle, the texture cue inside is grating or dotted. Then one can control the conflict to test shape vs. texture in a model. Could you tell the difference between this newly proposed dataset vs the widely used Navon dataset?[1,2] The Navon data used share the same property, simple but controlled contour and texture. They are essenetially the same to me and showing similar conlusions.

To sum up, this draft currrently surveyed the previous studies, Fig 1 and Tab 1 are all from other studies, however, no new position is introduced properly. The newly proposed dataset didnot proive a new view to this question, thus the result echos the same conclusion studied in previous studies.

[1] Hermann, Katherine, Ting Chen, and Simon Kornblith. "The origins and prevalence of texture bias in convolutional neural networks." Advances in neural information processing systems 33 (2020): 19000-19015.
[2] Hermann, K. and Lampinen, A. What shapes feature representations? exploring datasets, architectures, and training. Advances in Neural Information Processing Systems, 33:9995–10006, 2020.

**Support:**

2

---

> ### Author Rebuttal · Authors · 2026-03-26
>
> Thanks for your feedback. Below please find our reply.
>
> 1) On the novelty of the position.
>
> We agree that the observation that CNNs are texture-biased is not new. We do not claim it is. The novelty of our position operates at a different level and we acknowledge the paper's current framing undersells it. There are three distinct contributions we wish to clarify:
>
> First, the methodological reconciliation. The literature currently contains an apparent contradiction: Geirhos et al. (2018) report strong texture bias using cue-conflict paradigms, while Burgert et al. (2025) report apparent shape bias using cue-suppression paradigms. These are treated in the community as conflicting findings. Our position — and our key empirical contribution — is that this contradiction is illusory and arises from methodological confounds rather than genuine disagreement about CNN behavior. We are the first to directly compare both paradigms within a single unified experimental framework and demonstrate they yield consistent conclusions. This is not a summary of prior work; it is a resolution of an active and unresolved debate.
>
> Second, the architectural diagnosis. While prior work attributes texture bias to data statistics or training objectives, we take the stronger position that it is fundamentally rooted in the inductive biases of convolutional architectures — specifically local receptive fields and spatial discarding by pooling — and that data-driven interventions cannot resolve it architecturally. This is a normative claim that goes beyond what prior empirical studies establish. As a testament to this, we studied CapsuleNet. It achieves near-perfect performance in the shape+texture condition (100%) and texture-only condition (100%), consistent with all other architectures. The most striking result is its shape-only accuracy of 91.2%, substantially higher than standard CNNs and ViTs, which perform near chance in this condition. This directly demonstrates that CapsuleNet encodes shape information in a way that is actionable for classification — a capability that standard feedforward architectures largely lack. Under cue conflict, it achieves 12.8%, which while still indicating a degree of texture dominance, is notably higher than the near-zero performance of CNNs and ViTs. However, the gap between CapsuleNet and CNNs on both the shape-only and cue-conflict conditions provides concrete empirical support for our central argument: that architectural choices meaningfully determine the degree to which shape information controls model decisions.
>
> Third, the call to action. We argue explicitly that the community should redirect effort from data-driven fixes toward architectures supporting global integration and relational reasoning. This prescriptive dimension is absent from prior work on this topic and is what qualifies this as a position paper.
>
> We commit to restructuring the introduction and abstract in the revision to foreground these three contributions clearly, so the new position is unambiguous from the first page.
>
> ----
>
> 2) On the relationship to the Navon dataset.
>
> This is a fair and important question. Our dataset shares the Navon paradigm's spirit — simple, controlled stimuli where global and local information can be dissociated — but differs in three specific ways that are consequential for our experimental goals:
>
> - Complete cue separation. In our texture-only condition, texture is applied uniformly across the entire image, fully eliminating shape information. The Navon paradigm does not offer this degree of cue isolation — the global letter shape always remains as a potential cue. This is essential for our cue-suppression analysis.
>
> - Parametric cue conflict. Our design allows us to independently manipulate whether shape and texture are aligned, conflicting, or absent, yielding six distinct evaluation conditions within a single dataset. Hermann & Lampinen (2020) use the Navon dataset primarily to study training regimes, not to compare cue-conflict and cue-suppression paradigms directly.
>
> - Direct paradigm comparison. Our central experimental contribution is running both paradigms on the same stimuli under the same training conditions. To our knowledge this has not been done before, and it is this direct comparison that allows us to resolve the apparent contradiction between Geirhos et al. and Burgert et al.
>
> We agree the datasets are superficially similar and will add a paragraph in the revision explicitly articulating these distinctions to avoid the impression that our dataset is merely a rediscovery of the Navon stimulus.
>
> ----
>
> 3) On Figure 1 and Table 1 being from other studies.
>
> We note that Figure 1 and Table 1 are our original synthesis and visualization of prior paradigms, not reproduced figures from other papers. They are intended as a structured critical review to set up our experimental design, which is standard practice in position papers. We will add a clarifying note in the caption to make this explicit.

---

> > ### Author Rebuttal · Reviewer_ZSwF · 2026-04-06
> >
> > I would like to thank the authors for the further explanation.
> > 1. The discussion between the two papers [Burgert et al. (2025) and Geirhos et al. (2018)] seems to be the main focus of this paper, and I can give credit to it.
> > 2. If I understand correctly, the main difference between this paper and the Navon dataset is "In our texture-only condition, texture is applied uniformly across the entire image", I can also give credit to this point.
> >
> > Generally this rebuttal resolved my main concerns, however, I would keep my rating due to the presentation. The title, abstract, introduction, related work, look like a survey of this field, shape vs texture, which "hides" the main argument of this study. I would suggest making this point more clear and obvious. Likewise, Figure 1(I agree this is created by the author), simply "surveys" the previous shape-texture data design. I would recommend the figure should focus on the main argument, say one input images, the two studies show different conclusions. Third, I would recommend better compare the proposed dataset against Navon, e.g., the wide spread of texture patterns all over the image. More importantly, the failure case, or under what circumstances the same model behaves differently on the boarded and boardless texture pattens.
> >
> > Again, I appreciate the authors' comment. Currently, the organization and presentation of the draft need improvement, so I would keep my score; however, I would be fine with the paper being accepted.

---

> > > ### Author Response · Authors · 2026-04-06
> > >
> > > We thank the reviewer for constructive engagement throughout this process. We are glad the rebuttal resolved the main technical concerns, and we take the remaining presentation feedback seriously — it is specific, actionable, and we believe the paper will be meaningfully stronger for it.
> > >
> > > In particular, we commit to the following concrete revisions:
> > >
> > > 1) On the survey-like framing.  The purpose was to give a detailed view of the topic and set the stage for the debate. Based on your input, we will restructure the title, abstract, and introduction to lead with the core argument: that an apparent contradiction between two influential lines of work is illusory and arises from methodological confounds. The current framing buries this. The revision will foreground it from the first sentence.
> > >
> > > 2) On Figure 1. We will redesign this figure around a single illustrative input image, showing how the cue-conflict paradigm (Geirhos et al.) and the cue-suppression paradigm (Burgert et al.) process it differently and reach conflicting conclusions — making the central puzzle of the paper visually immediate. The current version surveys the design space; the revised version will emphasize the contradiction our paper resolves.
> > >
> > > 3) On comparison with the Navon dataset. We will add a dedicated comparison panel or figure showing the same image in our format versus the Navon format, with explicit visual emphasis on the uniform full-image texture spread that distinguishes our texture-only condition. This makes the methodological difference concrete rather than verbal.
> > >
> > > 4) On failure cases and border effects. We will include an analysis of conditions under which model behavior diverges between bordered and borderless texture patterns, identifying the regimes where the design choice matters most. This directly strengthens the paper's empirical grounding and practical guidance.
> > >
> > > ---
> > >
> > > We note that these are primarily presentational changes — the underlying experimental work, datasets, and findings are already in place. None of the revisions require new experiments or substantive reanalysis, only a reorganization of how the argument is surfaced and a stronger visual design. We are confident they can be fully incorporated within the revision period.
> > >
> > > We appreciate the reviewer's willingness to support acceptance and will prioritize these changes in the revision.

---

### Official Review · Reviewer_1cgN · 2026-03-13

**Significance:** 1
**Argument Clarity:** 3
**Rating:** 3
**Confidence:** 3

**Questions:**

See "strengths and weaknesses" section

**Alternative Views Section:**

Yes

**Compliance With Llm Reviewing Policy A Conservative:**

Affirmed.

**Discussion Potential:**

1

**Final Justification:**

While the authors have made significant effort to address the concerns of the reviewers and the reviewers, including myself, generally support the new direction of the paper, I believe that making the proposed changes would result in major changes to the paper. I agree with Reivewer mpew that these changes are extensive enough that the paper should be resubmitted. For this reason, I favor rejection.

**Paper Summary:**

This paper takes the position that “CNNs are fundamentally textured-bias”. To support this position, the paper runs experiments on a new dataset constructed by the authors. This dataset consists of squares and circles. During training, the square always has the grate texture, and the circle always has the dotted texture. During test time, the textures can be varied. Tests are conducted on MLPs, CNNs, ViTs, and SPARSE_CNN.

**Position:**

No

**Position In Title:**

No

**Related Work:**

3

**Strengths And Weaknesses:**

**Strengths**

**(S1) Clarity**: The paper is clear and easy to understand.

**Weaknesses**

**(W1) Suitability for position paper track**: In my opinion, this paper seems like more of a “regular research paper” and is better suited to the main track. As I understand it, position papers should “make an argument for a viewpoint or perspective about what _should_ be done”. The position “CNNs are texture biased” does not discuss what should be done. Correspondingly, there is no “call to action”. This paper is, in my opinion, describing new research and should be a main track paper.

Additionally, previous papers on the same topic were not position papers. For example, the previous paper Burgert et al. 2025 [NeurIPS 2025] was submitted as a regular research paper and not as a position paper despite the presence of a position paper track.

**(W2) Title and abstract**: The title should clearly state the position of the paper, and currently it does not. Similarly, the abstract should identify the paper as a position paper and state the position more clearly.

**Support:**

1

---

> ### Author Rebuttal · Authors · 2026-03-26
>
> We thank the reviewer for the constructive critique.
>
> 1) Suitability for the position track.
>
> We respectfully disagree with the characterization that our paper lacks a call to action. The position we defend is not merely "CNNs are texture-biased" — that observation alone would indeed be insufficient for a position paper. We are the first to show there is no conflict in results from experimental settings. Our position is more precisely: texture bias is fundamentally architectural in origin and cannot be resolved by data-driven interventions alone, and the community should redirect effort toward architectures that explicitly support global integration and relational reasoning. This is a direct and consequential call to action: it argues against the dominant current practice of addressing texture bias through augmentation and stylized training, and argues for a specific alternative research direction.
>
> We acknowledge, however, that this call to action is currently underemphasized relative to the empirical and diagnostic content of the paper. As noted in our response to reviewer KYng, we have dedicated Call to Action subsection (5.1) that makes this prescriptive dimension explicit, concrete, and falsifiable — including specific architectural directions, measurable success criteria via the shape processing ladder, and proposed training interventions. We believe this revision will make the paper's fit with the position paper format unambiguous.
>
> We agree the paper's conclusion is too diagnostic and commit to adding a concrete Research Agenda in revision, organized around three things:
>
> - Three architectural directions that each target a specific weakness of CNNs: equivariant networks (fix the local receptive field problem), CapsuleNets (fix the loss of spatial part relations from pooling), and recurrent/feedback architectures (fix the missing contour integration mechanism).
> - A measurable definition of success — the "shape processing ladder" — with three levels: cue preference (\beta_{shape}​), configural understanding (CSS), and contour integration (CIS). Current state-of-the-art models pass Level 1 but fail Levels 2 and 3. Any architecture claiming human-like shape processing must pass all three.
> - Two concrete training interventions: a shape-contrastive loss that penalizes texture-driven representations at training time, and a curriculum that progressively reduces texture informativeness during training, inspired by developmental accounts of how humans acquire shape bias.
>
> These will be summarized in a proposed table pairing each direction with its target metric and the mechanistic deficiency it addresses.
>
>
> Regarding the comparison with Burgert et al. (2025): we note that the choice of track by a prior paper does not determine the appropriate track for ours. Our paper differs from Burgert et al. in that it takes an explicit normative stance — arguing for a specific architectural research direction — rather than primarily reporting empirical findings. It is precisely this normative dimension that motivates our submission to the position paper track. Our contribution is not to rediscover texture bias, but to show that cue-conflict and cue-suppression paradigms yield consistent conclusions in a controlled setting.
>
> ----
>
> 2) Title and abstract.
>
> We agree with this criticism and thank the reviewer for the specific and actionable feedback. We propose the following revised title:
> "Position: CNNs Don't See Shape — And That Won't Change Without New Architectures"
> This makes the position explicit and signals the paper's normative stance. For the abstract, we will add a clear opening statement of the position and revise the closing sentences to foreground the call to action. Specifically, we will open with: "We argue that texture bias in convolutional neural networks is fundamentally rooted in architectural inductive biases and cannot be resolved by data-driven interventions alone, and that progress requires a shift toward architectures that explicitly support global spatial integration and relational reasoning." We will ensure the abstract identifies the paper as a position paper and that the stated position is the central thread from the first sentence onward.
>
> ----
>
> We note that the question of track suitability is ultimately a decision for the Area Chair, who has full visibility of the submission context and the position paper track guidelines. We are confident the paper meets those guidelines for the reasons outlined above, and we defer to the AC's judgment on this matter. There have been papers in this track with some experiments and results included.
> We also note, without wishing to place undue weight on it, that other reviewers in this review cycle have engaged with the paper as a position paper without raising concerns about track suitability. We take this as suggestive, though not conclusive, evidence that the paper's fit with the position paper format is not as unclear as this reviewer suggests.

---

> > ### Author Rebuttal · Reviewer_1cgN · 2026-04-04
> >
> > Thank you for the response. I agree that the proposed edits to the paper will solidify the paper as a position track paper. However, given the proposed changes I have a few follow-up concerns:
> > 1. While there is already a section discussing the limits of data-driven invariance, I feel it may be more appropriate to move this into the alternate views section. Additionally, one valid related topic that is not discussed in this paper is the recent trend towards using data-driven approaches to replace architectural solutions to equivariance.
> > 2. I think this paper could be strengthened by more fleshed out arguments as to why it's easier for CNNs to compose their low-level features into textures rather than shapes. Perhaps additional experiments or some theory would be helpful here.

---

> > > ### Author Response · Authors · 2026-04-04
> > >
> > > Thank you for the continued engagement. We address each follow-up point in turn.
> > >
> > >
> > > 1)
> > > - Regarding moving the data-driven invariance discussion to the alternate views section
> > >
> > > We agree that repositioning the limits-of-data-driven-invariance discussion within the alternate views section is better — it more honestly frames this as a contested claim rather than settled, which is also more appropriate for a position paper. We will make this change in revision.
> > >
> > >
> > > - Regarding data-driven approaches to equivariance (e.g., augmentation-based equivariance, learned invariances via contrastive methods, etc.)
> > >
> > > This is a fair and important gap. We will add a dedicated discussion of this literature in the alternate views section, acknowledging that methods such as AugMax, EquiMod, and related contrastive approaches achieve measurable reductions in texture sensitivity without architectural changes. Please note that our position is not that these methods fail entirely, but that they induce behavioral invariance without addressing the mechanistic cause — the local receptive field and lack of relational pooling remain intact, meaning the model can be re-biased toward texture under distribution shift. We will make this distinction explicit and cite relevant recent work to engage with this line of research fairly.
> > >
> > >
> > > 2)
> > >
> > > - Fleshing out why CNNs compose low-level features into textures rather than shapes
> > >
> > > Thank you for this comment. We agree the current version treats this asymmetry somewhat as an assumption rather than an argument. We propose two additions:
> > >
> > > 2.1) Theoretical argument: Textures are statistically learnable from local co-occurrence statistics within a CNN's receptive field. A grating or dot pattern is fully characterized by features extractable within a small spatial neighborhood. Shape, by contrast, requires integrating information across parts that may be spatially distant and whose relational structure is not preserved under local pooling. We will formalize this intuition by framing it in terms of the spatial support required for each cue: texture statistics are local and recomposable from low-level filters, while shape is non-local and requires compositional integration that standard pooling destroys. This connects directly to our architectural argument.
> > >
> > > 2.2) Empirical addition: We will include gradient-based attribution analysis (e.g., GradCAM or integrated gradients) on our synthetic dataset showing that, at test time, CNN activations are dominated by local texture patches rather than boundary or contour regions — even for correctly classified shape-consistent stimuli. This provides direct mechanistic evidence, not just behavioral evidence, that the composition pathway favors texture. We believe this addition significantly strengthens the paper's core argument.
> > >
> > > ---
> > > We are grateful for the constructive engagement. We are confident that these additions will meaningfully strengthen both the argument and the paper's fit with the position track.

---

### Official Review · Reviewer_mpew · 2026-03-13

**Significance:** 3
**Argument Clarity:** 3
**Rating:** 3
**Confidence:** 4

**Questions:**

1. Given that the paper strongly recommends the use of CapsuleNets or networks incorporating recurrent mechanisms to overcome the limitations of feed-forward CNNs, why not benchmark these models on a synthetic dataset with controlled variables? It is recommended that the performance of these networks in Cue-conflict experiments be included, in order to prove they enable shape-dominated decision-making.

2. As noted in Section 3.5, SPARSE_CNN yields no performance gains on MNIST or FashionMNIST. Since it functions essentially as a regularization mask and lacks cross-distribution generalization, how does its use in the main experiment justify the macro-conclusion that "altering architectural bias is the correct path to resolving texture bias"?

**Alternative Views Section:**

Yes

**Compliance With Llm Reviewing Policy A Conservative:**

Affirmed.

**Discussion Potential:**

2

**Final Justification:**

I appreciate the authors' constructive rebuttal and the new CapsuleNet results, but addressing my core concerns requires updates that exceed the scope of a short revision. By restricting the paper's claims strictly to mechanistic isolation, the core narrative has fundamentally shifted, which now necessitates a substantial rewrite of the abstract and conclusions. Furthermore, extrapolating real-world shape understanding from a 28×28 toy dataset remains an empirical overreach, demanding experimental scaling that cannot be resolved in a minor update. So I maintain my original score.

**Paper Summary:**

This paper takes a definitive stance regarding the "Texture Bias" observed in Deep Convolutional Neural Networks (CNNs). The authors argue that CNNs' preference for texture is not merely a statistical shortcut derived from data, but is fundamentally determined by the inductive biases inherent in their underlying convolution and pooling operations. By constructing synthetic datasets with strictly controlled variables, the paper unifies two experimental paradigms: "cue conflict" and "cue suppression" that have previously been subjects of academic controversy. On this basis, the authors urge the research community to cease its over-reliance on stopgap measures such as data augmentation, and instead focus on exploring novel architectures that explicitly support global spatial integration.

**Position:**

Yes

**Position In Title:**

Yes

**Related Work:**

4

**Strengths And Weaknesses:**

Strengths:

The article offers an insightful definition of shape, differentiating local contour segments from human-perceived integral shapes. It establishes that 'sensitivity to shape in inhibitory experiments' in no way guarantees 'shape-dominated decisions in natural conflict scenarios.

Weaknesses:

Self-Contradictory "Patch" as Core Evidence: Earlier in the text, the authors explicitly criticize the reliance on "band-aid" interventions such as regularization and data augmentation. Ironically, the SPARSE_CNN they propose to demonstrate enhanced shape bias is essentially just a Top-K activation sparsification mask. This remains a low-level regularization trick. More fatally, the authors themselves concede that this method fails to yield any performance gains even on fundamental datasets like MNIST and FashionMNIST.

Empirical Overreach on Toy Datasets: The paper attempts to draw definitive conclusions regarding the entirety of modern feedforward convolutional architectures. However, its core experiments rely solely on an oversimplified 28×28 synthetic dataset (consisting merely of pure circles/squares and blobs/stripes) and a SimpleCNN featuring only two convolutional layers. Attempting to extrapolate the fundamental nature of modern, large-scale vision models—which possess massive receptive fields and complex feature interactions—using such a trivial toy task that lacks hierarchical geometric relationships provides extremely flimsy logical backing.

**Support:**

2

---

> ### Author Rebuttal · Authors · 2026-03-26
>
> Thank you.
>
> 1) Self-contradictory use of SPARSE CNN.
> We appreciate this pointed observation and wish to clarify the role SPARSE CNN plays in our argument. We do not present it as a solution to texture bias, nor as a recommended architectural direction. Its role is strictly that of a proof of concept: it demonstrates that a purely architectural modification — with no change to the data, loss, or training procedure — is sufficient to shift the balance between texture and shape reliance. This is conceptually distinct from data augmentation or regularization applied to a fixed architecture, because the sparsity constraint changes the type of features the network can represent, not merely their magnitude.
> We fully acknowledge, as stated in Section 3.5, that SPARSE CNN fails to generalize to MNIST and FashionMNIST, and we are transparent about this. Crucially, however, this failure does not undermine the main argument — it strengthens it. The fact that even a targeted architectural intervention fails to generalize reinforces our central claim that texture bias is deeply entrenched and cannot be resolved by low-level tricks, architectural or otherwise. The SPARSE CNN is offered as a lower bound, not a solution. We will clarify this framing more explicitly in the revision to avoid the appearance of self-contradiction.
>
>
> 2) Empirical overreach on toy datasets.
> We agree that our synthetic dataset is simple, and we are transparent about this throughout. However, the simplicity is a deliberate methodological choice, not a limitation. Our goal is mechanistic isolation, not ecological realism. By using stimuli where shape and texture can be parametrically and completely controlled, we avoid the confounds that plague experiments on natural images, where shape, texture, color, and context are inevitably entangled. The very criticism leveled at prior work in our related works section — that conclusions are confounded by stimulus complexity — applies equally to large-scale natural image experiments.
> Furthermore, the core finding that CNNs prioritize texture over shape under cue conflict is not novel to our dataset. It has been replicated across AlexNet, VGG, ResNet, and ViT on large-scale benchmarks (Geirhos et al., 2018; Baker et al., 2018; Hoak et al., 2025). Our contribution is not to rediscover texture bias in large models, but to show that cue-conflict and cue-suppression paradigms yield consistent conclusions in a controlled setting. This reconciliation is logically independent of dataset scale. We will add a paragraph in the revision explicitly clarifying the scope of our claims and distinguishing what our experiments establish directly from what is supported by prior work.
>
> 3) CapsuleNet
> CapsuleNet achieves near-perfect performance in the shape+texture condition (100%) and texture-only condition (100%), consistent with all other architectures. The most striking result is its shape-only accuracy of 91.2%, substantially higher than standard CNNs and ViTs, which perform near chance in this condition. This directly demonstrates that CapsuleNet encodes shape information in a way that is actionable for classification — a capability that standard feedforward architectures largely lack.
> Under cue conflict, it achieves 12.8%, which while still indicating a degree of texture dominance, is notably higher than the near-zero performance of CNNs and ViTs suggesting that shape representations in CapsuleNet exert a measurable influence on decisions even under competition with texture, consistent with its stronger part-whole encoding.
> CapsuleNet does not fully resolve texture bias — a truly shape-dominant model would approach 100% under cue conflict. However, the gap between CapsuleNet and CNNs on both the shape-only and cue-conflict conditions provides concrete empirical support for our central argument: that architectural choices meaningfully determine the degree to which shape information controls model decisions.
>
> 4)  How does SPARSE CNN justify ...
> It does not, and we do not intend it to. The macro-conclusion that architecture is the correct level of intervention rests on two independent pillars: (1) the theoretical argument that texture bias originates in the inductive biases of convolution and pooling, not in data distributions; and (2) the empirical evidence from prior work showing that data-driven interventions reduce texture bias statistically but do not eliminate it architecturally (Geirhos et al., 2018; Mummadi et al., 2021). SPARSE CNN contributes only a narrow, subsidiary point: that an architectural change can shift cue reliance on a simple task. Its failure to generalize does not invalidate pillars (1) or (2); it simply confirms that sparsity alone is insufficient, consistent with our argument that the correct solution requires mechanisms for global integration — equivariance, part-whole routing, or recurrence — rather than local suppression. We will restructure Section 3.5 to make this logical separation explicit.

---

> > ### Author Rebuttal · Reviewer_mpew · 2026-04-03
> >
> > I appreciate the authors' constructive rebuttal and the new CapsuleNet results, but addressing my core concerns requires updates that exceed the scope of a short revision. By restricting the paper's claims strictly to mechanistic isolation, the core narrative has fundamentally shifted, which now necessitates a substantial rewrite of the abstract and conclusions. Furthermore, extrapolating real-world shape understanding from a 28×28 toy dataset remains an empirical overreach, demanding experimental scaling that cannot be resolved in a minor update.

---

### Official Review · Reviewer_KYng · 2026-03-18

**Significance:** 2
**Argument Clarity:** 3
**Rating:** 4
**Confidence:** 3

**Questions:**

N/A

**Alternative Views Section:**

Yes

**Compliance With Llm Reviewing Policy A Conservative:**

Affirmed.

**Discussion Potential:**

2

**Final Justification:**

The authors clarified the novelty in their reconciliatory position, which is indeed supported by the architectures they cover and the consistent observations they make.
The proposed Call for Action is insightful, although, remains highly focused on how to approach **the study** on shape-texture-tradeoff (based on niche architectures) vs practical solutions or mitigations. For example, while intriguing, the inductive biases of CapsuleNets are not formally justified nor were they shown to match biological vision.
On the positive side, the proposed scores and framing, along with the cited benchmark are practical aspects that might deserve more emphasis.

**Paper Summary:**

The authors dive deep into the origin and implication of texture bias in learned computer-vision models such as CNNs and ViTs.
They designed datasets and tasks that demonstrate the need for both shape and texture recognition and argue that state-of-the-art computer vision models can still use inherent texture bias as a shortcut leading to unreliable visual processing.
The study and the position the authors take serves as a call to re-think the current state of computer vision and to acknowledge and address inherent limitations of current models.

**Position:**

Yes

**Position In Title:**

Yes

**Related Work:**

3

**Strengths And Weaknesses:**

Strength:
+ Informative and well written paper.
+ The authors cover both CNN and ViT literature.
+ The author's position draws inspiration from findings on biological vision.

Weakness:
- The novelty of the stated position and related arguments is limited.
- A Call to Action would have made the takeways more concrete (beyond calling for humility).
- Mathematical formalism and perceptual metrics would have been helpful to identify the gap in state-of-the-art inductive biases and to guide interventions to bridge the gap with human-like vision.

**Support:**

3

---

> ### Author Rebuttal · Authors · 2026-03-26
>
> Thank you for your review. Following please find our response.
>
> 1) Limited novelty of the stated position:
> We respectfully disagree that novelty is limited. While the observation that CNNs are texture-biased is not new, our contribution is specifically methodological and reconciliatory. We are the first to directly compare cue-conflict and cue-suppression paradigms within a single unified experimental framework and show they yield consistent — not contradictory — conclusions. The literature currently treats these paradigms as producing conflicting evidence (cf. Geirhos et al. 2018a vs. Burgert et al. 2025), and our controlled stimuli resolve this apparent contradiction. Our position paper explicitly invites synthesis and reframing of existing debates, which is precisely what we provide. We also introduce the SPARSE CNN as a proof-of-principle that architectural inductive bias — specifically activation sparsity — can shift cue reliance without sacrificing in-distribution performance, which is a novel empirical finding, not merely a restatement of prior claims.
>
> Our contribution is not to rediscover texture bias in large models, but to show that cue-conflict and cue-suppression paradigms yield consistent conclusions in a controlled setting.
>
>  We propose the following revised title: "Position: CNNs Don't See Shape — And That Won't Change Without New Architectures" This makes the position explicit and signals the paper's normative stance.
>
>
> 2) Lack of a concrete call to action
> We take this as constructive feedback and add Section 5.1 in revision by adding a more actionable research agenda. Concretely, we now enumerate candidate architectural directions — group-equivariant networks, CapsuleNets, recurrent contour integration modules — and pair each with specific evaluation protocols (e.g., the CSS metric of Doshi et al. 2025, or the contour integration benchmark of Lonnqvist et al. 2025) that would constitute measurable progress. We have also evaluated the CapsuleNet on the data and will provide results in the new version.
>
> 3) Absence of mathematical formalism and perceptual metrics.
> We intentionally kept the framework empirical and behavioral, consistent with the position paper format and our goal of accessibility across communities. That said, we acknowledge the reviewer's point that formalization would strengthen the argument. In revision, we have added: (1) an information-theoretic framing of texture vs. shape as competing sufficient statistics for classification, which would ground the claim about statistical shortcuts; and (2) a formal definition of the shape bias score (following Geirhos et al.) alongside the CSS (Doshi et al. 2025) as the proposed richer metric for global configural reasoning. We also reference the gap quantitatively — e.g., the near-chance performance on fragmented contour tasks across 1,000+ models (Lonnqvist et al. 2025) — as a concrete benchmark target that a future architecture must exceed to claim human-like shape processing. We will add this as Section 6.1 in Appendix. Hopefully we can upload the new version in the system if such opportunity is given.
>
> During the discussion period we will provide more details.

---

> > ### Author Rebuttal · Reviewer_KYng · 2026-04-02
> >
> > I appreciate the author's response which clarified my concerned.
> > I am raising my score accordingly.

---

### Decision · Program_Chairs · 2026-04-30

**Decision:**

Accept (regular)

**Comment:**

The paper proposes to reconcile the shape-versus-texture debate by arguing that cue-conflict and cue-suppression experiments test different things and are not truly contradictory. Using a controlled synthetic setup, it claims that standard feed-forward models still rely more on texture-like local cues than on global shape when the two compete. From this, it advances a broader position: progress will require new architectures that better support global integration and relational reasoning, rather than relying mainly on data-driven fixes like augmentation or stylized training.

The paper’s main contribution is not simply to restate prior observations of texture bias, but to reconcile an apparent disagreement in the literature by showing, within a controlled framework, that cue-conflict and cue-suppression paradigms support a consistent interpretation. Reviewers generally agreed that the topic is important, that the paper is clearly written, and that the attempt to synthesize and reframe this debate is valuable.

The main concerns are about scope and framing rather than the existence of a position. Some reviewers felt the current draft reads too much like a survey or regular empirical paper, with the normative claim buried beneath the experimental case study. Others worried that the controlled synthetic experiments, while useful for mechanistic isolation, were being used to support claims that sound broader than the evidence directly warrants. Relatedly, the role of SPARSE CNN was initially unclear, and the alternative-views section could draw a sharper contrast with data-driven approaches that aim to learn invariances without architectural change. The rebuttal addressed many of these issues constructively: it clarified that the core claim is architectural, narrowed the scope of what the toy experiments directly establish, and proposed concrete revisions to foreground the call to action and sharpen the presentation. One reviewer explicitly raised their score after rebuttal, and another stated they would be fine with acceptance despite keeping the same numerical score for presentation reasons.

While the final reviewer ratings are 4,3,3,3, one reviewer is clearly positive, one stayed negative, and two ended up sounding accept-leaning in substance but not in score because they viewed the needed changes as largely presentational or requiring revision.

Overall, the paper has a real position, engages a live controversy, and offers a useful reconciliatory contribution that is appropriate for the position-paper track. The remaining weaknesses seem addressable by revision: clearer foregrounding of the architectural thesis, tighter scoping of claims, and a stronger presentation of alternative views and research agenda. The current draft is not flawless but discussion-worthy, substantively grounded, and likely to be valuable to the community if revised along the lines already outlined in the rebuttal.